# ISG15 Is a Novel Regulator of Lipid Metabolism during *Vaccinia Virus* Infection

Manuel Albert,[a] Jesús Vázquez,[b] Juan M. Falcón-Pérez,[c] María A. Balboa,[d,e] Marc Liesa,[f,g] Jesús Balsinde,[d,e] Susana Guerra[a]

aDepartment of Preventive Medicine, Public Health and Microbiology, Universidad Autónoma de Madrid, Madrid, Spain
bCentro Nacional de Investigaciones Cardiovasculares (CNIC-ISCIII), Madrid, Spain
cCIC-bioGUNE—Centro de Investigación Cooperativa en Biociencias, Vizcaya, Spain
dConsejo Superior de Investigaciones Científicas (CSIC), Instituto de Biología y Genética Molecular, Valladolid, Spain
eCentro de Investigación Biomédica en Red de Diabetes y Enfermedades Metabólicas Asociadas (CIBERDEM), Instituto de Salud Carlos III, Madrid, Spain
fDepartment of Medicine, Endocrinology, David Geffen School of Medicine at UCLA, Los Angeles, California, USA
gInstitut de Biologia Molecular de Barcelona, IBMB, CSIC, Barcelona, Spain

**ABSTRACT** Interferon-stimulated gene 15 (ISG15) is a 15-kDa ubiquitin-like modifier that binds to target proteins in a process termed ISGylation. ISG15, first described as an antiviral molecule against many viruses, participates in numerous cellular processes, from immune modulation to the regulation of genome stability. Interestingly, the role of ISG15 as a regulator of cell metabolism has recently gained strength. We previously described ISG15 as a regulator of mitochondrial functions in bone marrow-derived macrophages (BMDMs) in the context of *Vaccinia virus* (VACV) infection. Here, we demonstrate that ISG15 regulates lipid metabolism in BMDMs and that ISG15 is necessary to modulate the impact of VACV infection on lipid metabolism. We show that $Isg15^{-/-}$ BMDMs demonstrate alterations in the levels of several key proteins of lipid metabolism that result in differences in the lipid profile compared with $Isg15^{+/+}$ (wild-type [WT]) BMDMs. Specifically, $Isg15^{-/-}$ BMDMs present reduced levels of neutral lipids, reflected by decreased lipid droplet number. These alterations are linked to increased levels of lipases and are independent of enhanced fatty acid oxidation (FAO). Moreover, we demonstrate that VACV causes a dysregulation in the proteomes of BMDMs and alterations in the lipid content of these cells, which appear exacerbated in $Isg15^{-/-}$ BMDMs. Such metabolic changes are likely caused by increased expression of the metabolic regulators peroxisome proliferator-activated receptor-$\gamma$ (PPAR$\gamma$) and PPAR$\gamma$ coactivator-1$\alpha$ (PGC-1$\alpha$). In summary, our results highlight that ISG15 controls BMDM lipid metabolism during viral infections, suggesting that ISG15 is an important host factor to restrain VACV impact on cell metabolism.

**IMPORTANCE** The functions of ISG15 are continuously expanding, and growing evidence supports its role as a relevant modulator of cell metabolism. In this work, we highlight how the absence of ISG15 impacts macrophage lipid metabolism in the context of viral infections and how poxviruses modulate metabolism to ensure successful replication. Our results open the door to new advances in the comprehension of macrophage immunometabolism and the interaction between VACV and the host.

**KEYWORDS** host-pathogen interactions, innate immunity, interferons

The interferon-stimulated gene 15 (*Isg15*) encodes the protein ISG15, a 15-kDa small ubiquitin-like modifier that binds proteins through a process termed ISGylation (1). ISGylation targets *de novo*-synthesized proteins in a cotranslational manner (2, 3) and is a reversible process thanks to the action of the ISG15-specific protease USP18 (4). ISG15 can also act as a free molecule either inside or outside the cell, modifying

Address correspondence to Susana Guerra, susana.guerra@uam.es.

The authors declare no conflict of interest.

protein stability and functions, and acts as a cytokine with immunomodulatory effects (5, 6).

The diversity of functions performed by ISG15 is continuously expanding in part due to its versatile mode of action. Apart from its well-established antiviral activity (7), in recent years, research has linked ISG15 to numerous pathways, including cancer (8), genome stability (9), intracellular trafficking (10), autophagy (11–13), vascular remodeling (14), and metabolism (15–19). The role of ISG15 as a regulator of cell metabolism has recently gained strength, especially after the discovery of the close relationship between ISG15 and mitochondria (20). We previously demonstrated that ISG15 regulates mitochondrial respiration, dynamics, and mitophagy in bone marrow-derived macrophages (BMDMs) (16), which was further confirmed for other cell types (15). ISG15 was shown to participate in the reprogramming of liver metabolism during *Listeria monocytogenes* and *Coxsackievirus* B3 infection (17, 19). In addition, it was recently shown that in adipocytes, signaling through the interferon regulatory factor 3 (IRF3)/ISG15 axis results in the inhibition of glycolytic enzymes through ISGylation, strongly reducing oxygen consumption and thermogenic capacity (18). Of note, Waqas and colleagues (21) recently demonstrated that macrophages from patients who suffer congenital ISG15 deficiency display metabolic alterations that affect pathways such as mitochondrial respiration, glycolysis, amino acid metabolism, and redox balance, among others, highlighting the relevance of ISG15 as a regulator of cell metabolism also in humans.

The host's metabolic state is a determining factor of how the host responds to viral infections. Viruses lack their own metabolism and, hence, depend on the host's metabolic processes to complete their replicative cycles. However, virus-host interactions are not passive, as viruses encode proteins that modulate host metabolism for their own benefit (22). In this sense, poxviruses possess a clear advantage, as they encode around 200 different proteins, of which a high percentage serves to modulate host immune responses (23–25). *Vaccinia virus* (VACV) is the prototype virus of the *Orthopoxvirus* genus of the *Poxviridae* family. VACV is the live vaccine used for the eradication of smallpox and one of the most extensively studied poxviruses (26). As with every other virus, VACV depends on the cellular metabolic state to complete its infectious cycle and has different strategies to manipulate cell metabolism. VACV has been shown to induce a pseudohypoxic state early after infection through the stabilization of hypoxia-inducible factor-1$\alpha$ (HIF-1$\alpha$) by the VACV protein C16 (27). This shifts glycolytic flux toward lactate production and induces an increase in glutamine consumption to provide carbons to the tricarboxylic acid (TCA) cycle (28, 29). This allows the generation of citrate to synthesize fatty acids (FAs), which are oxidized in the mitochondrion to fuel oxidative phosphorylation (OXPHOS)-linked ATP production (30). In this line, it was recently shown that the VACV protein C11, which is a viral growth factor (VGF), is responsible for enhancing the production of TCA cycle intermediates during VACV infection (31). Lipids also play a relevant role during VACV infection, apart from their function as an energy source. Cholesterol content on membrane lipid rafts is important for fusion of VACV with cell membranes, which determines virus entry and egress (32). In addition, several VACV proteins require FA acylation (e.g., palmitoylation and myristoylation) for proper function (33, 34). Additionally, as an enveloped virus, alterations in cellular lipid content impact VACV membrane generation and composition (35). Therefore, the efficiency of VACV infection is expected to be dependent on the mitochondrial and lipid metabolism of host cells.

Considering the role of ISG15 as a regulator of macrophage metabolism during VACV infection (16), in this study, we explored how ISG15 controls macrophage lipid metabolism and how its presence or absence affects how VACV modulates lipid metabolism during infection. Our results broaden the repertoire of functions performed by ISG15 and demonstrate the relevance of ISG15 in the modulation of VACV-host interactions.

## RESULTS

**ISG15 modulates the levels of proteins involved in lipid metabolism independently of gene expression.** To determine the role of ISG15 in lipid metabolism, we performed a quantitative proteomic analysis of wild-type (WT) and *Isg15*$^{-/-}$ BMDMs.

Considering that the levels of ISG15 and ISGylated proteins are practically undetectable in unstimulated BMDMs (16), we treated our cells with type I interferon (IFN; 500 U/mL) 16 h before the proteomic analysis. The proteomic analysis identified and annotated 7,154 proteins. Of the total, 2,199 proteins were considered significantly differentially expressed between genotypes, as their $Zq$ values (standardized $\log_2$ [$Isg15^{-/-}$/WT] ratio at the protein level) were higher or lower than +2 and −2, respectively. Nine hundred and eighty-three proteins were significantly upregulated ($Zq \geq +2$), while 1,216 proteins were significantly downregulated ($Zq \leq -2$). These proteins were subjected to bioinformatic analyses using the Ingenuity Pathway Analysis (IPA) software. We mined proteomic data sets focusing on key mitochondrial proteins regulating lipid metabolism and found that several mitochondrial proteins involved in FA metabolism, mainly fatty acid oxidation (FAO), were differentially expressed in $Isg15^{-/-}$ BMDMs (Table 1, bold). Interestingly, most of the proteins identified were upregulated, in line with our previous results in which we observed a general upregulation of mitochondrial proteins in $Isg15^{-/-}$ BMDMs (16). Moreover, according to the IPA canonical pathway analysis, many of the proteins detected belonged to the FAO pathway. In addition, we were excited to discover that not only mitochondrial proteins involved in FA metabolism were differentially expressed between genotypes but also nonmitochondrial enzymes that participate in diverse aspects of lipid metabolism, such as lipid synthesis, lipid hydrolysis, and lipid transport and storage (Table 1). Interestingly, proteins involved in lipid hydrolysis (e.g., neutral cholesterol ester hydrolase 1 [NCEH1], monoacylglycerol lipase [MGLL], lysosomal acid lipase/cholesterol ester hydrolase [LIPA], and lipid droplet-associated hydrolase [LDAH]) were upregulated in $Isg15^{-/-}$ BMDMs, whereas critical enzymes in lipid synthesis, such as farnesyl pyrophosphate synthase (FDPS), acetyl-coenzyme A (acetyl-CoA) carboxylase A (ACACA), and fatty acid synthase (FASN), were strongly downregulated in these cells. These results suggested that the absence of ISG15 shifts lipid metabolism toward lipid hydrolysis and oxidation and, therefore, an imbalance between lipid anabolic and catabolic processes.

To evaluate whether the differences observed at the protein level were due to variations in gene expression, we analyzed the mRNA levels of several lipid metabolism genes by quantitative reverse transcription-PCR (RT-qPCR). We selected genes involved in lipid hydrolysis (*Nceh1*), lipid synthesis (*Fasn*, *Hmgcr*, and *Hmgcs*), mitochondrial FAO (*Cpt1a*, *Acads*, *Acadm*, and *Acadl*), and lipid transport and storage (*Cd36*, *Soat1*, *Plin2*, and *Abcg1*). Surprisingly, none of the genes examined showed significant differences in mRNA levels between genotypes (Fig. S1 in the supplemental material), indicating that differences in the levels of these proteins were independent of gene expression. It is well known that gene expression does not always correlate with protein abundance due to posttranscriptional and posttranslational mechanisms involved in the regulation of protein homeostasis and turnover (36), including ISGylation (37), which might explain these results.

**The macrophage lipid profile is altered in the absence of ISG15.** To confirm whether the changes in protein content of enzymes regulating lipid metabolism resulted in a change in their function, we performed a lipidomic analysis of type I IFN-treated WT and $Isg15^{-/-}$ BMDMs to define the effects of *Isg15* deletion on the content of different lipid species. The lipidomic analysis detected a total of 226 lipids, which were subjected to univariate and multivariate data analyses.

Multivariate data analysis with all the samples and pooled samples was performed to determine the quality and reproducibility of the measurements, which were reported as high. Also, this analysis revealed a clear clustering of samples according to the genetic background, indicating substantial differences in the lipid profiles between genotypes (Fig. S2A, top). Metabolites responsible for this separation were triacylglycerols (TAGs), sphingomyelins (SMs), ceramides (Cer), and cholesterol esters (CEs), which were decreased in $Isg15^{-/-}$ BMDMs, and ethanolamine glycerophospholipids (PEs), which were increased in these cells (Fig. S2A, bottom).

Univariate data analysis was performed to calculate percent changes, and unpaired Student's $t$ test $P$ values (or Welch's $t$ test when necessary) were used for the

**TABLE 1** Differences in lipid metabolism protein abundances between IFN-treated $Isg15^{-/-}$ and WT BMDMs

| Pathway | Protein ID | Gene[a] | Description | Std $\log_2$ (FC)[b] ($Isg15^{-/-}$ versus WT) |
|---|---|---|---|---|
| Fatty acid oxidation | Q8BMS1 | **Hadha** | Trifunctional enzyme subunit alpha, mitochondrial | 11.68 |
| | P50544 | **Acadvl** | Very long-chain-specific acyl-CoA dehydrogenase, mitochondrial | 10.62 |
| | Q99JY0 | **Hadhb** | Trifunctional enzyme subunit beta, mitochondrial | 9.52 |
| | Q9EPL9 | **Acox3** | Peroxisomal acyl-coenzyme A oxidase 3 | 8.97 |
| | P97742 | **Cpt1a** | Carnitine O-palmitoyltransferase 1, liver isoform | 8.54 |
| | Q8BH95 | **Echs1** | Enoyl-CoA hydratase, mitochondrial | 8.31 |
| | Q07417 | **Acads** | Short-chain-specific acyl-CoA dehydrogenase, mitochondrial | 7.62 |
| | P51174 | **Acadl** | Long-chain-specific acyl-CoA dehydrogenase, mitochondrial | 6.74 |
| | Q8QZT1 | **Acat1** | Acetyl-CoA acetyltransferase, mitochondrial | 6.14 |
| | P42125 | **Eci1** | Enoyl-CoA delta isomerase 1, mitochondrial | 6.00 |
| | Q9R0H0 | Acox1 | Peroxisomal acyl-coenzyme A oxidase 1 | 5.34 |
| | P45952 | **Acadm** | Medium-chain-specific acyl-CoA dehydrogenase, mitochondrial | 5.03 |
| | P52825 | **Cpt2** | Carnitine O-palmitoyltransferase 2, mitochondrial | 4.69 |
| | Q60759 | **Gcdh** | Glutaryl-CoA dehydrogenase, mitochondrial | 4.47 |
| | Q9DBL1 | **Acadsb** | Short/branched-chain-specific acyl-CoA dehydrogenase, mitochondrial | 4.41 |
| | Q9Z2Z6 | **Slc25a20** | Mitochondrial carnitine/acylcarnitine carrier protein | 4.14 |
| | P41216 | **Acsl1** | Long-chain fatty acid-CoA ligase 1 | 3.38 |
| | P32020 | **Scp2** | Sterol carrier protein 2 | 3.24 |
| | Q9DI5 | **Mcee** | Methylmalonyl-CoA epimerase, mitochondrial | −3.02 |
| | P16332 | **Mmut** | Methylmalonyl-CoA mutase, mitochondrial | −6.67 |
| | Q8CAY6 | Acat2 | Acetyl-CoA acetyltransferase, cytosolic | −8.05 |
| Lipid synthesis | Q9D517 | Agpat3 | 1-Acyl-sn-glycerol-3-phosphate acyltransferase gamma | 4.87 |
| | Q8BYI6 | Lpcat2 | Lysophosphatidylcholine acyltransferase 2 | 2.88 |
| | Q8CHK3 | Mboat7 | Lysophospholipid acyltransferase 7 | 2.87 |
| | Q8BHF7 | Pgs1 | CDP-diacylglycerol-glycerol-3-phosphate 3-phosphatidyltransferase, mitochondrial | 2.73 |
| | O35083 | Agpat1 | 1-Acyl-sn-glycerol-3-phosphate acyltransferase alpha | 2.60 |
| | Q91YX5 | Lpgat1 | Acyl-CoA:lysophosphatidylglycerol acyltransferase 1 | 2.31 |
| | Q8K3K7 | Agpat2 | 1-Acyl-sn-glycerol-3-phosphate acyltransferase beta | 2.10 |
| | Q5SWU9 | Acaca | Acetyl-CoA carboxylase 1 | −2.20 |
| | Q3UJQ2 | Hmgcs1 | Hydroxymethylglutaryl-CoA synthase C domain-containing protein | −3.06 |
| | Q920E5 | Fdps | Farnesyl pyrophosphate synthase | −10.60 |
| | P19096 | Fasn | Fatty acid synthase | −15.65 |
| Lipid hydrolysis | Q8BLF1 | Nceh1 | Neutral cholesterol ester hydrolase 1 | 43.15 |
| | Q8VCI0 | Plbd1 | Phospholipase B-like 1 | 19.16 |
| | O35678 | Mgll | Monoglyceride lipase | 16.29 |
| | Q9WV54 | Asah1 | Acid ceramidase | 14.26 |
| | Q91WC9 | Daglb | Diacylglycerol lipase-beta | 5.10 |
| | Q9Z0M5 | Lipa | Lysosomal acid lipase/cholesteryl ester hydrolase | 4.26 |
| | Q8VEB4 | Pla2g15 | Phospholipase A2 group XV | 4.04 |
| | Q8BVA5 | Ldah | Lipid droplet-associated hydrolase | 3.51 |
| | Q80Y98 | Ddhd2 | Phospholipase DDHD2 | −2.27 |
| Lipid transport and storage | P08226 | Apoe | Apolipoprotein E | 17.93 |
| | Q91ZX7 | Lrp1 | Prolow-density lipoprotein receptor-related protein 1 | 12.93 |
| | Q08857 | Cd36 | Platelet glycoprotein 4 | 3.86 |
| | P41233 | Abca1 | Phospholipid-transporting ATPase ABCA1 | 3.60 |
| | Q00623 | Apoa1 | Apolipoprotein A-I | 3.46 |
| | Q61009 | Scarb1 | Scavenger receptor class B member 1 | −3.38 |
| | P11404 | Fabp3 | Fatty acid-binding protein, heart | −3.47 |
| | P48410 | Abcd1 | ATP-binding cassette subfamily D member 1 | −3.61 |
| | P43883 | Plin2 | Perilipin-2 | −4.59 |
| | Q9DBG5 | Plin3 | Perilipin-3 | −9.87 |

[a]Mitochondrial proteins are highlighted in bold, and targets of ISGylation are underlined.
[b]FC, fold change; Std, standard.

comparisons between $Isg15^{-/-}$ and WT BMDMs. A heatmap was generated to help visualize the results (Fig. 1A). A detailed analysis of the comparison between genotypes showed that 112 of 226 lipids were significantly altered in $Isg15^{-/-}$ BMDMs compared with WT cells. Our attention was caught by the fact that almost the whole profile of

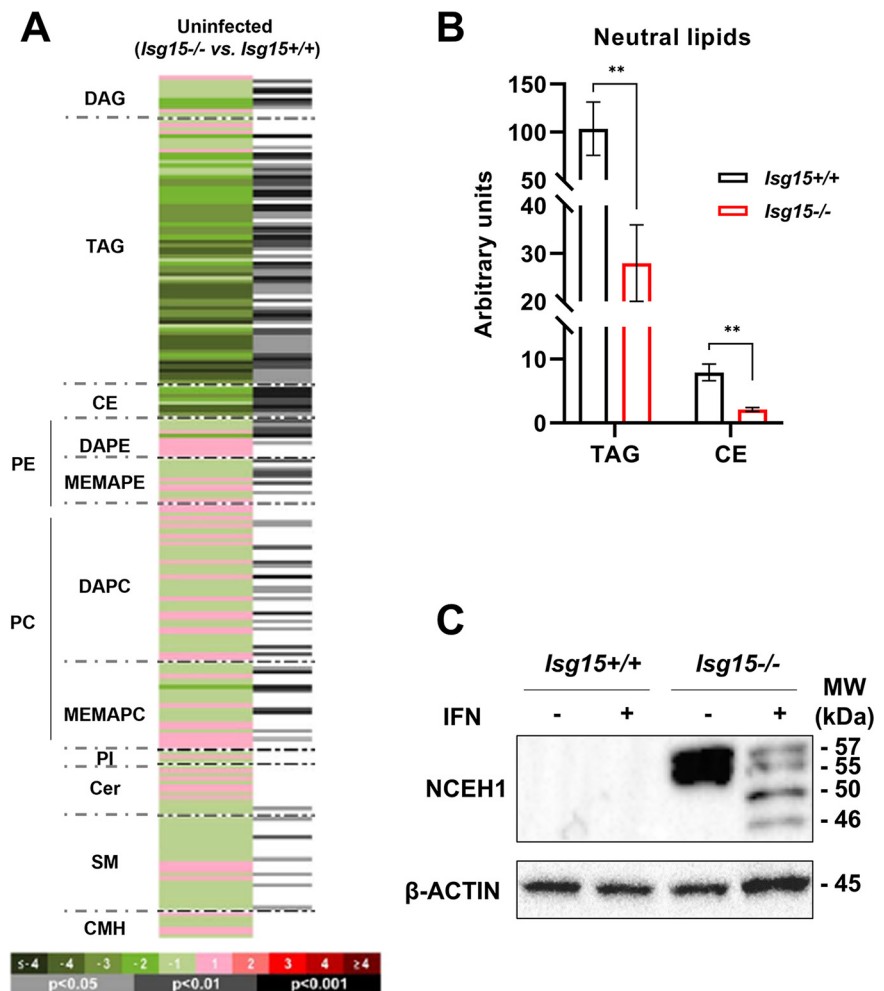

**FIG 1** Analysis of the lipid profile of *Isg15*⁻/⁻ and WT BMDMs. (A and B). Lipidomic analysis of IFN-treated *Isg15*⁻/⁻ and WT BMDMs. BMDMs from 4 mice of each genotype were subjected to UHPLC-MS-based metabolomics analysis. (A) The heatmap shows the log₂ fold change (*Isg15*⁻/⁻ versus WT) of the 226 metabolites analyzed, together with the *P* value obtained in the appropriate statistical analysis. Darker green and red colors indicate higher drops or elevations of the metabolites represented, respectively. Gray lines correspond to significant fold change values of individual metabolites, where darker gray colors indicate higher significances. Also, metabolites are represented in order according to their carbon number and unsaturation degree of acyl changes. CMH, monohexosylceramides; DAPE, diacylglycerophosphoethanolamines; MEMAPE, 1-ether, 2-acylglycerophosphoethanolamines; DAPC, diacyl-glycerophosphocholines; MEMAPC, 1-ether, 2-acylglycerophosphocholines; PI, phosphatidylinositols. (B) *Isg15*⁻/⁻ BMDMs show lower levels of neutral lipids. The levels of TAGs and CEs obtained in the lipidomic analysis are shown. Mean ± standard deviation (SD) of 4 biological replicates is represented; *, *P* < 0.05; **, *P* < 0.01; ***, *P* <0.001. (C) *Isg15*⁻/⁻ BMDMs show increased expression of NCEH1. Total protein extracts (25 μg) of untreated and IFN-treated *Isg15*⁻/⁻ and WT BMDMs were subjected to 7.5% SDS-PAGE and resolved by Western blotting. Antibodies against NCEH1 and β-actin (control) were used (Table S1 in the supplemental material). Molecular weights (MW) are indicated in kDa.

neutral lipids (NLs; TAGs and CEs) decreased in *Isg15*⁻/⁻ BMDMs (Fig. 1A and B). NLs play a wide range of functions in the cell, from FA storage in lipid droplets (LDs) to the generation of lipid second messengers involved in numerous signaling pathways (38), thus being key molecules in the regulation of immune responses (39). Considering the decrease in NLs observed in *Isg15*⁻/⁻ cells and the relevance of LDs in innate immunity, we focused our study on these lipid species. However, it is also important to remark that other lipid species, mainly diacylglycerols (DAGs) and phospholipids (e.g., PEs and choline glycerophospholipids [PCs]), were also altered in *Isg15*⁻/⁻ BMDMs (Fig. 1A), indicating that ISG15 is necessary to maintain homeostasis of a wide variety of lipid molecules.

It was remarkable that CEs were the most affected NLs in $Isg15^{-/-}$ BMDMs. Interestingly, NCEH1 was one of the most highly upregulated proteins in $Isg15^{-/-}$ BMDMs (Table 1). NCEH1 is a neutral CE hydrolase that resides in the endoplasmic reticulum (ER) and was shown to determine macrophage cholesterol homeostasis (40, 41). As we showed previously, the high levels of NCEH1 were independent of gene expression (Fig. S1); therefore, we analyzed NCEH1 levels by Western blotting. Surprisingly, NCEH1 was only detected in $Isg15^{-/-}$ BMDMs, in line with the strong upregulation observed in the proteomic analysis (Fig. 1C). The increase in NCEH1 protein content was independent of IFN treatment, indicating that its overexpression is the result of the absence of ISG15, even under basal conditions. However, IFN caused changes in the band pattern, shifting from an intense double band around 50 kDa to a less intense doublet, an additional 48-kDa band, and the native 46-kDa band (Fig. 1C). These observations suggested that NCEH1 is processed or modified in response to type I IFN, although the cause and outcomes of these modifications are still to be determined. Altogether, our data demonstrate that the absence of ISG15 results in a reduction in the NL content (mainly CEs) in BMDMs, likely due to an increase in the levels of NL hydrolases.

**ISG15 regulates lipid droplet homeostasis in BMDMs.** Given the marked decrease in NLs detected in the lipidomic analysis, we evaluated the status of the lipid droplet (LD) pool in type I IFN-treated BMDMs. NLs were stained with the fluorescent dye boron-dipyrromethene (BODIPY) 493/503 to visualize LDs, which were imaged by confocal microscopy. Macrophages of both genotypes showed few and small LDs, and no clear differences were detected between genotypes (Fig. 2A, top). Therefore, to increase LD numbers and thus facilitate the analysis, LD synthesis was induced by treating cells with 100 $\mu$M oleic acid (OA) for 24 h. As expected, the LD content significantly increased in response to OA (Fig. 2A, bottom). To determine whether the LD population differed between genotypes, LDs were quantified, and the BODIPY-stained cell area was calculated. The number of LDs was notably reduced in IFN-treated $Isg15^{-/-}$ BMDMs compared with WT cells, although these differences were not statistically significant. In lipid-loaded cells, however, we detected a significant decrease in LD number in $Isg15^{-/-}$ BMDMs (Fig. 2B, left). As well, the lower BODIPY-stained area in $Isg15^{-/-}$ BMDMs further supported a reduction in NL and LD content in these cells (Fig. 2B, right). In addition, consistent with these results, our proteomic analysis reported a marked decrease in the LD-specific markers perilipin 2 (PLIN2) and PLIN3 in $Isg15^{-/-}$ BMDMs (Table 1). Together, our results demonstrate that ISG15 is necessary for the stability of the LD pool in BMDMs.

**The reduced LD content of $Isg15^{-/-}$ BMDMs is not linked to increased FAO.** The reduction in NL levels and LDs observed in $Isg15^{-/-}$ BMDMs proposed three possible scenarios: increased lipid hydrolysis from LDs to fuel FAO, decreased FA esterification and LD formation, or a combination of the above two situations. Our results suggested that the first scenario was the most likely to happen, as the levels of enzymes involved in lipid esterification were similar between genotypes (data not shown), in contrast to an increase in FAO proteins and lipid hydrolases and a decrease in the LD proteins PLIN2 and PLIN3 in $Isg15^{-/-}$ BMDMs (Table 1).

To evaluate FA dynamics in these cells, we performed a pulse-chase analysis with BODIPY 558/568 C12 (Red C12), an orange-red fluorescent FA that acts as an 18-carbon FA analog and can be incorporated into LDs (42). Cells were labeled with 1 $\mu$M Red C12 in growth medium for 16 h. Then, labeling medium was replaced with fresh medium, and red fluorescence and LD numbers were analyzed by confocal microscopy in live cells at 0 and 3 h after medium replacement (Fig. 3A). Quantification of Red C12 fluorescence indicated that there were no significant differences in total Red C12 content between genotypes at any time point evaluated (Fig. 3B, left), although there was an unexpected increase in Red C12 fluorescence at 3 h. However, the subcellular distribution of Red C12 seemed to be different between genotypes. It was noticeable that WT BMDMs accumulated more Red C12 in LDs (Fig. 3A), further evidenced by quantification of LD numbers (Fig. 3B, right), which were reduced in $Isg15^{-/-}$ BMDMs at any time point. These results open the door to defects in FA esterification as the cause of

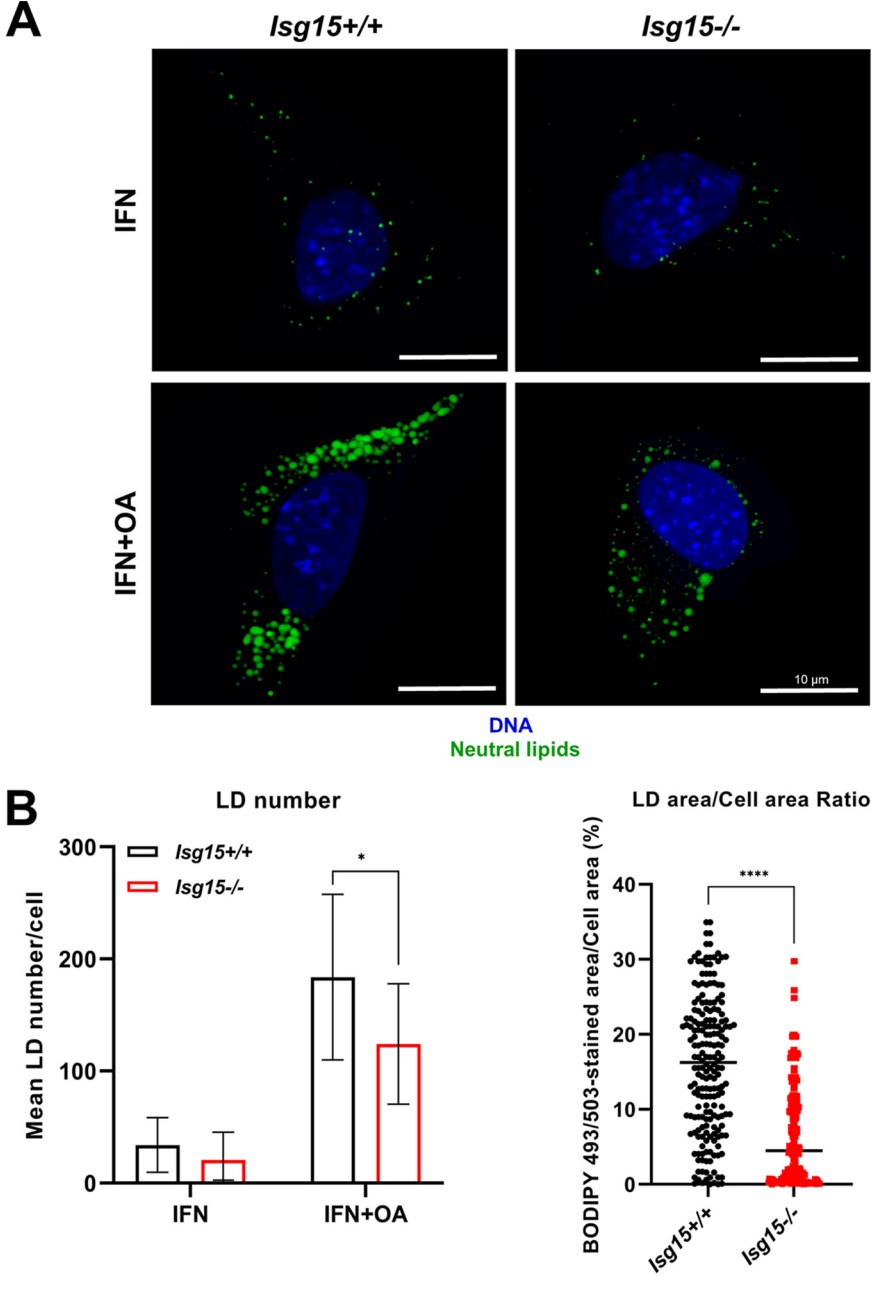

**FIG 2** Analysis of the LD content of *Isg15*$^{-/-}$ and WT BMDMs. (A and B) *Isg15*$^{-/-}$ BMDMs show less LDs than WT BMDMs. (A) *Isg15*$^{-/-}$ and WT BMDMs were treated with type I IFN (500 U/mL) alone or in combination with OA (100 $\mu$M) for 24 h. LDs were stained *in vivo* for 30 min at 37°C in a humidified incubator. Cells were fixed with 4% PFA and prepared for analysis by confocal microscopy. DNA was stained with DAPI. Microscopy analysis was performed in a Zeiss LSM 880 Airyscan superresolution microscope. Images were processed and analyzed with Aivia AI image analysis software. Representative images are shown. (B) LD number was calculated with Aivia AI image analysis software. Fifteen to 20 images of each condition were used to determine mean LD numbers (left). Two hundred to 250 images of each condition were used to determine the BODIPY 493/503-stained cell area (right). Mean $\pm$ SD is represented. A Student's *t* test was performed for the comparisons; *, $P < 0.05$; **, $P < 0.01$; ***, $P < 0.001$.

reduced LD number in these cells, as the differences in LD number were observed at 0 h since the medium was replaced. Nonetheless, unlike lipases, enzymes responsible for FA esterification were not altered in *Isg15*$^{-/-}$ BMDMs, which points to increased lipid hydrolysis and a faster lipid turnover as the cause of reduced LD numbers in *Isg15*$^{-/-}$ BMDMs.

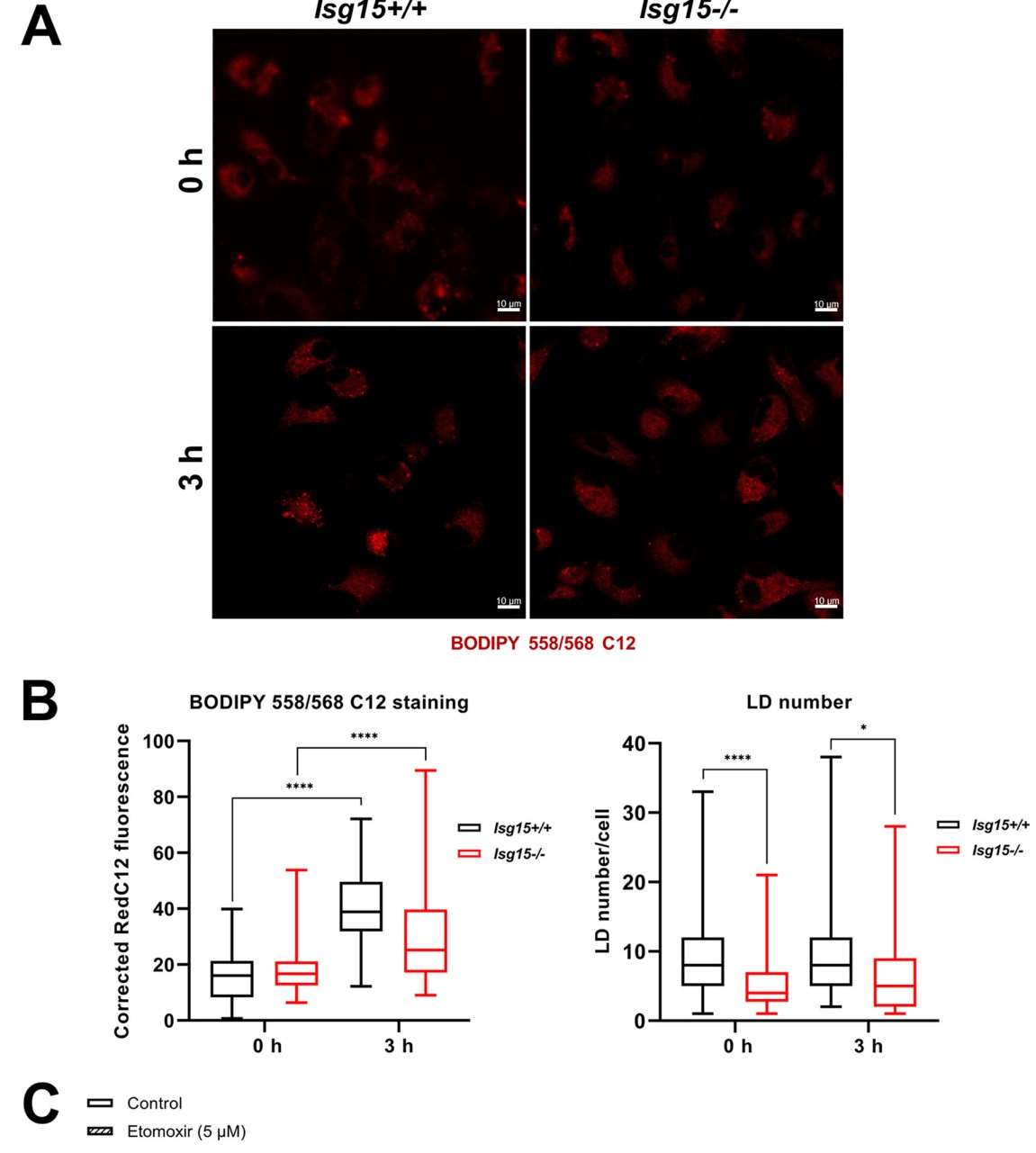

**FIG 3** Analysis of FA dynamics and mitochondrial FAO in $Isg15^{-/-}$ and WT BMDMs. (A and B) FAs are differently distributed between genotypes. (A) IFN-treated $Isg15^{-/-}$ and WT BMDMs were treated with 1 $\mu$M Red C12 for 18 h. Red C12 was removed, cells were washed, and fresh complete medium was

The fact that total Red C12 levels did not significantly differ between genotypes suggested that the destination of hydrolyzed lipids seemed not to be mitochondrial FAO. To assess whether there were differences in mitochondrial FAO between $Isg15^{-/-}$ and WT BMDMs, we measured basal and maximal oxygen consumption rate (OCR) and OXPHOS-linked ATP production in the presence of etomoxir (5 $\mu$M) using a Seahorse Biosciences XF96 extracellular flux analyzer platform. Etomoxir is a specific inhibitor of CPT1, the enzyme responsible for the import of long-chain FAs (LCFAs) into the mitochondrion for FAO (43). The inhibition of CPT1 by etomoxir treatment reduces FAO and, hence, results in a decrease of OCR and OXPHOS-derived ATP in cells that are using FAs as a carbon source. The respirometric analysis of IFN-treated BMDMs indicated that $Isg15^{-/-}$ BMDMs showed reduced basal and maximal OCR and lower levels of OXPHOS-derived ATP independent of etomoxir treatment (Fig. 3C). Surprisingly, etomoxir did not have any effect on OCR and ATP production neither in $Isg15^{-/-}$ BMDMs nor in WT BMDMs, as respiratory parameters were not altered compared with untreated cells. These results indicated that, despite the increase in FAO proteins observed in $Isg15^{-/-}$ BMDMs, FAO activity was not increased in these cells, which confirmed our previous results in which we observed a reduction in OXPHOS and mitochondrial ATP synthesis in $Isg15^{-/-}$ BMDMs (16). Altogether, these results suggested that lipolysis is stimulated in $Isg15^{-/-}$ BMDMs, although the generated free FAs are not oxidized in mitochondria.

**VACV infection and the absence of ISG15 have similar effects on macrophage metabolism.** Viruses are strictly dependent on the host's metabolic features; therefore, viruses have developed many strategies to modulate cell metabolism to ensure replication (22). Here, we explored the impact of VACV infection on macrophage metabolism. We first subjected the data from the proteomic analysis of IFN-treated, VACV-infected WT BMDMs to IPA canonical pathway analysis. Of the 3,144 proteins identified as significantly differentially expressed in response to infection, 1,698 proteins were significantly upregulated ($Zq \geq +2$), while 1,446 proteins were significantly downregulated ($Zq \leq -2$). These proteins were subjected to IPA canonical pathway analysis. The analysis reported mitochondria as the main target of VACV, as mitochondrial dysfunction and OXPHOS were the most altered pathways in VACV-infected BMDMs, together with pathways such as the TCA cycle, FAO, and reactive oxygen species (ROS) production (Fig. 4). These results are consistent with previous research that demonstrated that VACV manipulates mitochondrial metabolism during infection (16, 29–31, 44). The IPA canonical pathway analysis also reported alterations in several pathways of lipid metabolism, such as FAO, TAG biosynthesis, and cholesterol biosynthesis (Fig. 4). Hence, we also analyzed the levels of proteins involved in different aspects of lipid metabolism, such as FAO, lipid synthesis, lipid hydrolysis, and lipid transport and storage (Table 2). We found a general increase in protein levels in VACV-infected BMDMs; however, we detected the opposite behavior for some key proteins. For example, the levels of CPT1A, the main FAO enzyme, were reduced in response to VACV infection. Similarly, FASN levels were strongly decreased in VACV-infected BMDMs compared with in uninfected cells (Table 2). These observations might indicate that neither FAO nor FA synthesis (FAS) are increased during VACV infection of BMDMs, in contrast to what was observed for cell types such as BSC40 and human foreskin fibroblasts (HFF) (30, 31), suggesting that the metabolic response to VACV infection might be different between cell types. Interestingly, proteins involved in lipid uptake and storage, such as CD36, SOAT1, and PLIN2, were upregulated in VACV-infected BMDMs, pointing to

**FIG 3** Legend (Continued)

added. Red fluorescence was analyzed *in vivo* with an SP5 confocal microscope. Images at 0 and 3 h after medium replacement are shown. (B) Corrected Red C12 fluorescence was calculated with Fiji (left). LD number per cell was quantified in 30 or more different images from each condition (right). Mean $\pm$ SD is represented, and Student's *t* tests were performed for the comparisons. (C) Mitochondrial FAO is similar between genotypes. Mitochondrial OCR and OXPHOS-linked ATP levels of IFN-treated $Isg15^{-/-}$ and WT BMDMs were measured using a Seahorse Biosciences XF96 extracellular flux analyzer in the presence or not of etomoxir (5 $\mu$M). Mean $\pm$ SD of three biological replicates is represented. Student's *t* tests were performed for the comparisons; *, $P < 0.05$; **, $P < 0.01$; ***, $P < 0.001$.

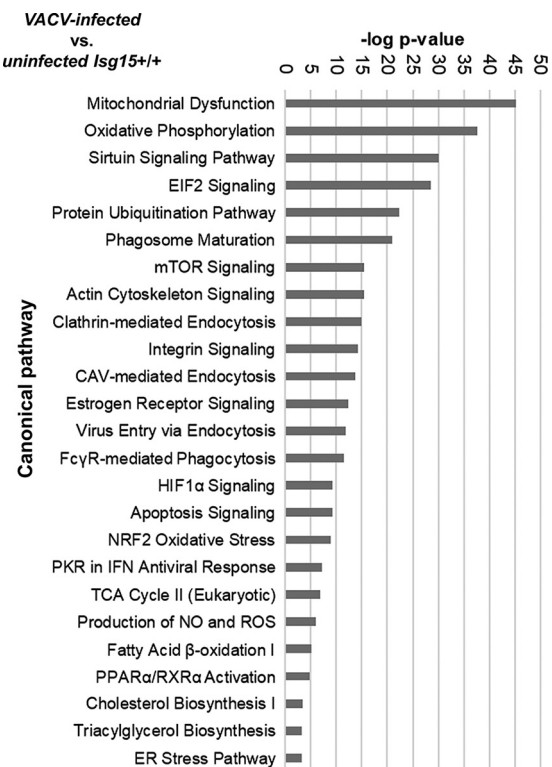

**FIG 4** Differentially expressed canonical pathways between IFN-treated VACV-infected and uninfected WT BMDMs. Data from the proteomic analysis of IFN-treated VACV-infected and uninfected WT BMDMs were subjected to the IPA canonical pathway analysis. Top differentially expressed pathways between genotypes are listed. Pathways are classified according to the *P* value of the comparison VACV-infected versus uninfected WT BMDMs; CAV, caveolin; NO, nitric oxide; EIF2, eukaryotic translation initiation factor 2; mTOR, mammalian target of rapamycin; PKR, RNA-activated protein kinase; ER, endoplasmic reticulum.

increased lipid influx and storage during VACV infection, a common feature of infected cells (45).

To evaluate whether the differences in protein levels were due to changes in gene expression, we analyzed mRNA levels of the same subset of lipid metabolism genes selected for uninfected BMDMs, which included genes involved in lipid hydrolysis (*Nceh1*), lipid synthesis (*Fasn*, *Hmgcr*, and *Hmgcs*), mitochondrial FAO (*Cpt1a*, *Acads*, *Acadm*, and *Acadl*), and lipid transport and storage (*Cd36*, *Soat1*, *Plin2*, and *Abcg1*). This analysis was also performed in VACV-infected *Isg15*$^{-/-}$ BMDMs to assess whether ISG15 has a role in the control of the expression of these genes during VACV infection. Significant differences were found for some genes in response to infection (Fig. 5A). *Cpt1a* mRNA levels were significantly increased in infected cells, a surprising result considering the strong downregulation observed in the proteomic analysis (Table 2), suggesting that posttranscriptional and/or posttranslational mechanisms might modulate the levels of CPT1A during VACV infection. The expression of *Plin2* and *Cd36* was also increased by VACV infection, in line with the results of the proteomic analysis, further supporting an increase in lipid uptake and storage during infection. It was also remarkable that the expression of several genes was notably increased in VACV-infected *Isg15*$^{-/-}$ BMDMs, as occurred with *Acads*, *Fasn*, *Hmgcr*, *Soat1*, and *Nceh1*, although the differences were not statistically significant.

An interesting observation was that the proteomic profile of VACV-infected WT BMDMs was highly similar to that of uninfected *Isg15*$^{-/-}$ BMDMs (Fig. 4) (16). VACV has been shown to impair the action of ISG15 via its protein E3 (46). Moreover, here we demonstrate that VACV causes a strong downregulation of *Isg15* expression (Fig. 5B, left) and a marked reduction of ISG15 and ISGylation levels in BMDMs (Fig. 5B, right). Based on these results, we hypothesize that the similarities between the proteomic profiles of uninfected *Isg15*$^{-/-}$

**TABLE 2** Differences in lipid metabolism protein abundances between VACV-infected and uninfected WT BMDMs

| Pathway | Protein ID | Gene[a] | Description | Std log$_2$ (FC)[b] (VACV versus uninfected Isg15$^{+/+}$) |
|---|---|---|---|---|
| Fatty acid oxidation | Q99MN9 | **Pccb** | Propionyl-CoA carboxylase beta chain, mitochondrial | 10.02 |
| | P51174 | **Acadl** | Long-chain-specific acyl-CoA dehydrogenase, mitochondrial | 8.64 |
| | Q91ZA3 | **Pcca** | Propionyl-CoA carboxylase alpha chain, mitochondrial | 8.63 |
| | Q9EPL9 | _Acox3_ | Peroxisomal acyl-coenzyme A oxidase 3 | 8.62 |
| | Q8QZT1 | **Acat1** | Acetyl-CoA acetyltransferase, mitochondrial | 7.71 |
| | Q9Z2Z6 | **Slc25a20** | Mitochondrial carnitine/acylcarnitine carrier protein | 6.67 |
| | Q9JHI5 | **Ivd** | Isovaleryl-CoA dehydrogenase, mitochondrial | 6.39 |
| | Q8BH95 | **_Echs1_** | Enoyl-CoA hydratase, mitochondrial | 5.58 |
| | Q8JZN5 | **Acad9** | Complex I assembly factor ACAD9, mitochondrial | 5.39 |
| | Q99JY0 | **Hadhb** | Trifunctional enzyme subunit beta, mitochondrial | 5.11 |
| | Q9CQ62 | **Decr1** | 2,4-Dienoyl-CoA reductase ([3E]-enoyl-CoA-producing), mitochondrial | 4.23 |
| | P50544 | **Acadvl** | Very long-chain-specific acyl-CoA dehydrogenase, mitochondrial | 3.55 |
| | Q9DBL1 | **Acadsb** | Short/branched-chain-specific acyl-CoA dehydrogenase, mitochondrial | 3.04 |
| | Q60759 | **Gcdh** | Glutaryl-CoA dehydrogenase, mitochondrial | 3.02 |
| | Q8K370 | **Acad10** | Acyl-CoA dehydrogenase family member 10 | 2.12 |
| | Q8CAY6 | **Acat2** | Acetyl-CoA acetyltransferase, cytosolic | −2.61 |
| | P97742 | **_Cpt1a_** | Carnitine O-palmitoyltransferase 1, liver isoform | −5.88 |
| | Q9CZS1 | **Aldh1b1** | Aldehyde dehydrogenase X, mitochondrial | −6.21 |
| | P32020 | _Scp2_ | Sterol carrier protein 2 | −6.86 |
| Lipid synthesis | O35083 | Agpat1 | 1-Acyl-sn-glycerol-3-phosphate acyltransferase alpha | 4.95 |
| | Q9D517 | Agpat3 | 1-Acyl-sn-glycerol-3-phosphate acyltransferase gamma | 4.62 |
| | Q8CHK3 | Mboat7 | Lysophospholipid acyltransferase 7 | 4.25 |
| | Q8BHF7 | Pgs1 | CDP-diacylglycerol-glycerol-3-phosphate 3-phosphatidyltransferase, mitochondrial | 4.21 |
| | Q8K3K7 | Agpat2 | 1-Acyl-sn-glycerol-3-phosphate acyltransferase beta | 3.29 |
| | Q91YX5 | Lpgat1 | Acyl-CoA:lysophosphatidylglycerol acyltransferase 1 | 2.47 |
| | Q8BYI6 | Lpcat2 | Lysophosphatidylcholine acyltransferase 2 | 2.44 |
| | Q8JZK9 | Hmgcs1 | Hydroxymethylglutaryl-CoA synthase C domain-containing protein | −2.34 |
| | Q920E5 | _Fdps_ | Farnesyl pyrophosphate synthase | −7.31 |
| | P19096 | _Fasn_ | Fatty acid synthase | −8.10 |
| Lipid hydrolysis | Q8VCI0 | Plbd1 | Phospholipase B-like 1 | 10.62 |
| | Q99LR1 | Abhd12 | Lysophosphatidylserine lipase ABHD12 | 9.89 |
| | Q3TCN2 | Plbd2 | Putative phospholipase B-like 2 | 7.85 |
| | Q9Z0M5 | Lipa | Lysosomal acid lipase/cholesteryl ester hydrolase | 7.12 |
| | Q91WC9 | Daglb | Diacylglycerol lipase-beta | 5.27 |
| | Q8BLF1 | Nceh1 | Neutral cholesterol ester hydrolase 1 | 4.72 |
| | Q9WV54 | Asah1 | Acid ceramidase | 4.12 |
| | Q8VEB4 | Pla2g15 | Phospholipase A2 group XV | 3.61 |
| | Q80YA3 | Ddhd1 | Phospholipase DDHD1 | 2.35 |
| | Q8BVA5 | Ldah | Lipid droplet-associated hydrolase | −2.57 |
| Lipid transport and storage | P08226 | Apoe | Apolipoprotein E | 13.09 |
| | Q61263 | Soat1 | Sterol O-acyltransferase 1 | 11.56 |
| | Q08857 | Cd36 | Platelet glycoprotein 4 | 10.39 |
| | P55302 | Lrpap1 | Alpha-2-macroglobulin receptor-associated protein | 9.58 |
| | Q91ZX7 | _Lrp1_ | Prolow-density lipoprotein receptor-related protein 1 | 7.90 |
| | P41233 | Abca1 | Phospholipid-transporting ATPase ABCA1 | 5.22 |
| | P48410 | Abcd1 | ATP-binding cassette subfamily D member 1 | 2.79 |
| | P43883 | _Plin2_ | Perilipin-2 | 2.65 |
| | Q61009 | Scarb1 | Scavenger receptor class B member 1 | 2.68 |
| | Q00623 | Apoa1 | Apolipoprotein A-I | 2.23 |
| | Q64343 | Abcg1 | ATP-binding cassette subfamily G member 1 | −3.01 |
| | Q3B7Z2 | Osbp | Oxysterol-binding protein 1 | −4.58 |
| | P11404 | Fabp3 | Fatty acid-binding protein, heart | −6.93 |

[a]Mitochondrial proteins are highlighted in bold, and targets of ISGylation are underlined.
[b]FC, fold change; Std, standard.

BMDMs and VACV-infected WT BMDMs might be due to reduced ISG15 levels in the context of viral infection or IFN stimulation. In line with this hypothesis, we previously demonstrated that uninfected Isg15$^{-/-}$ BMDMs and VACV-infected WT BMDMs show similar alterations in mitochondrial functions (16).

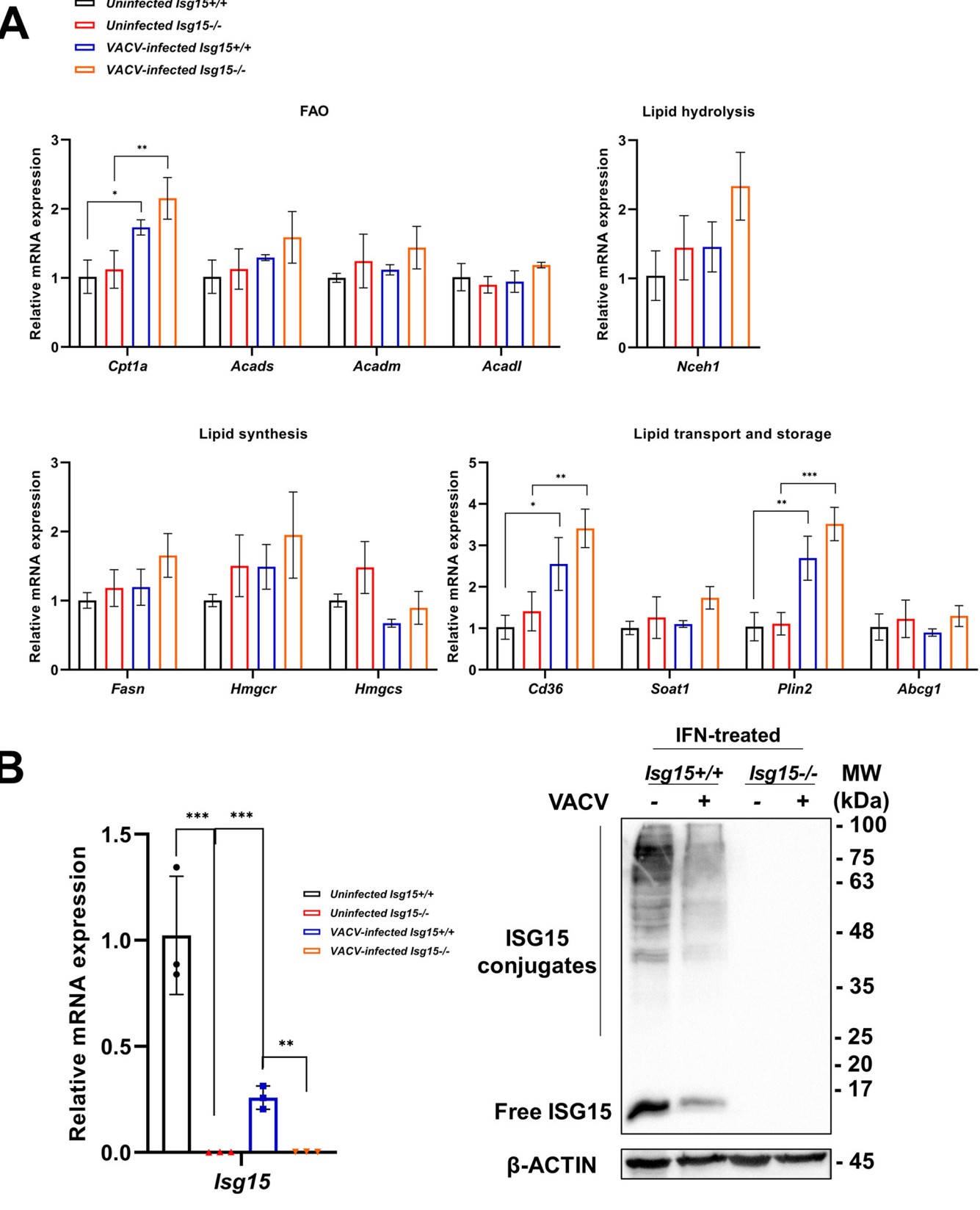

**FIG 5** Analysis of the expression of lipid metabolism genes and *Isg15* in uninfected and VACV-infected *Isg15*$^{-/-}$ and WT BMDMs. (A) VACV enhances the expression of key genes of FAO, lipid uptake, and lipid storage. mRNA levels of the indicated genes were analyzed by RT-qPCR in uninfected and VACV-infected *Isg15*$^{-/-}$ and WT BMDMs treated with IFN. Expression levels were normalized to *HPRT* mRNA levels. Mean ± SD of three biological replicates is represented. Two-way ANOVA and Tukey *post hoc* analyses were performed for the comparisons; *, $P < 0.05$; **, $P < 0.01$; ***, $P < 0.001$. (B) VACV reduces

**VACV alters the lipid profile of BMDMs.** To explore how proteomic changes induced by VACV infection affected the lipid homeostasis of BMDMs, we performed a lipidomic analysis of type I IFN-treated $Isg15^{-/-}$ and WT BMDMs infected with VACV wild-type Western Reserve strain (VACV WR; 1 PFU/cell) for 16 h.

Multivariate data analyses revealed a clear clustering of samples according to infection and genetic background. In the comparison between VACV-infected versus uninfected WT cells, DAGs, TAGs, CEs, Cer, and PCs were increased in samples of infected cells, whereas SMs and PEs were increased in samples of uninfected cells (Fig. S2B, top). Comparing samples of infected and uninfected $Isg15^{-/-}$ BMDMs, the main differences were found in CEs, Cer, PCs, several species of TAGs, and SMs, which were increased in infected cells, and DAGs and PEs, which were increased in uninfected cells (Fig. S2B, middle). Finally, the comparison between samples of infected $Isg15^{-/-}$ and WT BMDMs reported an increase of almost all the metabolites analyzed in WT cells (Fig. S2B, bottom), consistent with the results obtained for uninfected cells.

Univariate data analyses were performed for the comparisons between VACV-infected and uninfected cells of the same genotype and infected cells of the two different genotypes. As performed for uninfected cells, a heatmap was generated to help visualize the results (Fig. 6A). A detailed analysis of the comparison of VACV-infected and uninfected WT BMDMs reported that only 34 of 226 lipids were significantly altered in infected cells. Interestingly, TAGs and CEs were increased in infected cells, with significant differences in CEs (Fig. 6B, left), in line with our observations of upregulation of proteins involved in lipid uptake and flux (Table 2). The comparison between infected versus uninfected $Isg15^{-/-}$ BMDMs showed that only 50 of 226 lipids were significantly altered, highlighting a strong increase in almost the whole CE profile and several species of TAGs in infected cells (Fig. 6A and B, middle). Finally, comparing infected $Isg15^{-/-}$ and WT BMDMs, only seven lipids were significantly altered. Interestingly, significant differences were reported mostly for CE species, which were increased in infected $Isg15^{-/-}$ BMDMs (Fig. 6A and B, right).

Given the marked increase in NLs in response to VACV infection, we evaluated the status of the LD pool in VACV-infected BMDMs by confocal microscopy. We infected IFN-treated $Isg15^{-/-}$ and WT BMDMs with VACV WR (multiplicity of infection [MOI] of 1 for 16 h) and analyzed the LD content by confocal microscopy. On one hand, the reduction in LD number in $Isg15^{-/-}$ BMDMs compared with WT BMDMs was confirmed (Fig. 6C). On the other hand, VACV infection caused a significant increase in the number of LDs in WT BMDMs, whereas such an increase was not statistically significant in $Isg15^{-/-}$ BMDMs. Additionally, the differences in LD number between genotypes were still observed in the context of VACV infection. The increase in LD number in response to VACV infection is consistent with increased NL levels in infected cells, as indicated by the lipidomic analysis (Fig. 6A and B).

Altogether, our results show that VACV induces the accumulation of NLs and the formation of LDs in BMDMs and that ISG15 is necessary to restrain the effect of VACV on BMDM lipid metabolism.

**Absence of ISG15 and VACV infection alters the transcriptional program of lipid metabolism.** To better understand the possible origin of the observed metabolic changes, and with the aim to elucidate a mechanism by which ISG15 and VACV infection could be altering these processes, we subjected our proteomics data to the IPA upstream regulators analysis. This analysis allows the identification of regulators that can explain the observed differences in protein levels based on the number of known targets of each regulator present in the data set. In addition, the software predicts the

**FIG 5** Legend (Continued)
protein ISGylation. $Isg15^{-/-}$ and WT BMDMs were treated or not with type I IFN (500 U/mL for 16 h) and infected or not with VACV (1 PFU/cell for 16 h). Total RNA was extracted, and mRNA levels of $Isg15$ were analyzed by RT-qPCR (left). Total protein extracts (20 $\mu$g) were subjected to 10% SDS-PAGE and resolved by Western blotting (right). Antibodies against ISG15 and $\beta$-actin (control) were used (Table S1 in the supplemental material). Molecular weights (MW) are indicated in kDa. Mean $\pm$ SD of three biological replicates is represented. Two-way ANOVA and Tukey *post hoc* analyses were performed for the comparisons; *, $P < 0.05$; **, $P < 0.01$; ***, $P < 0.001$.

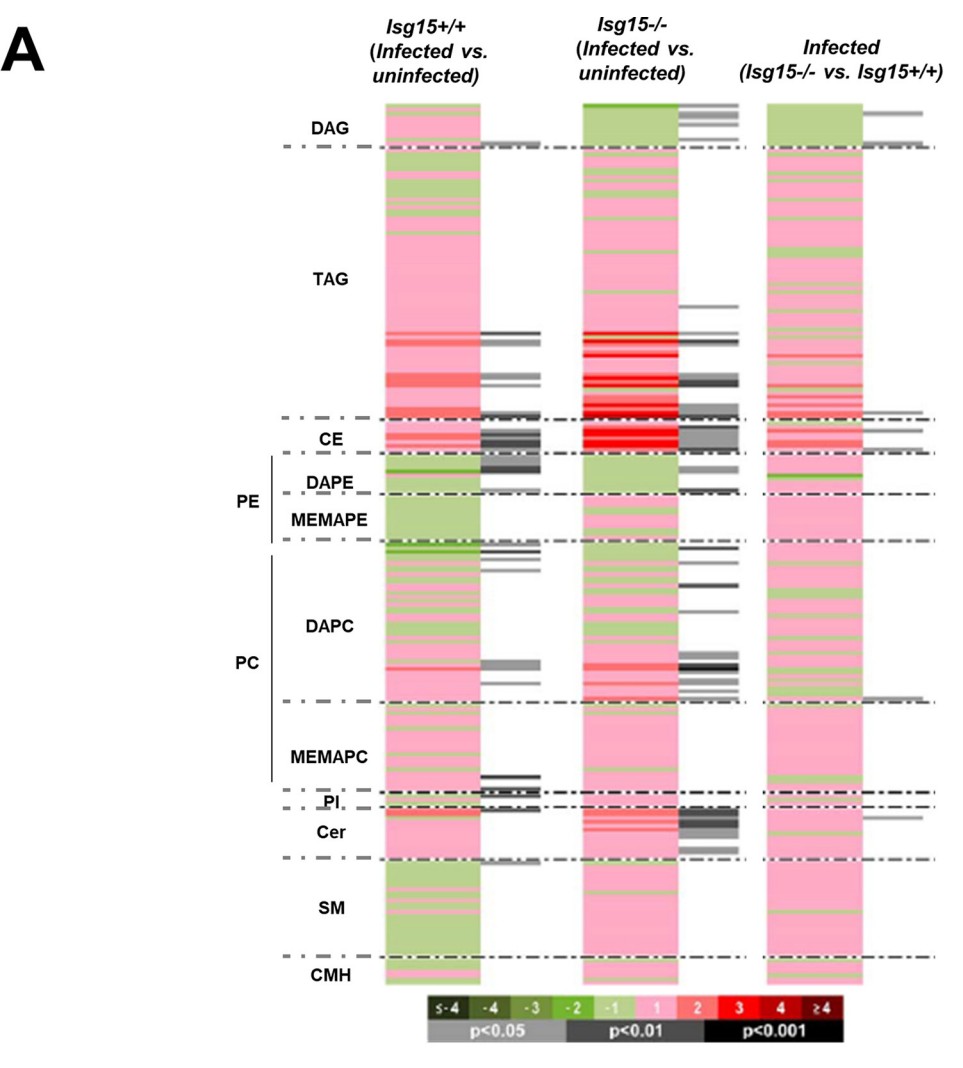

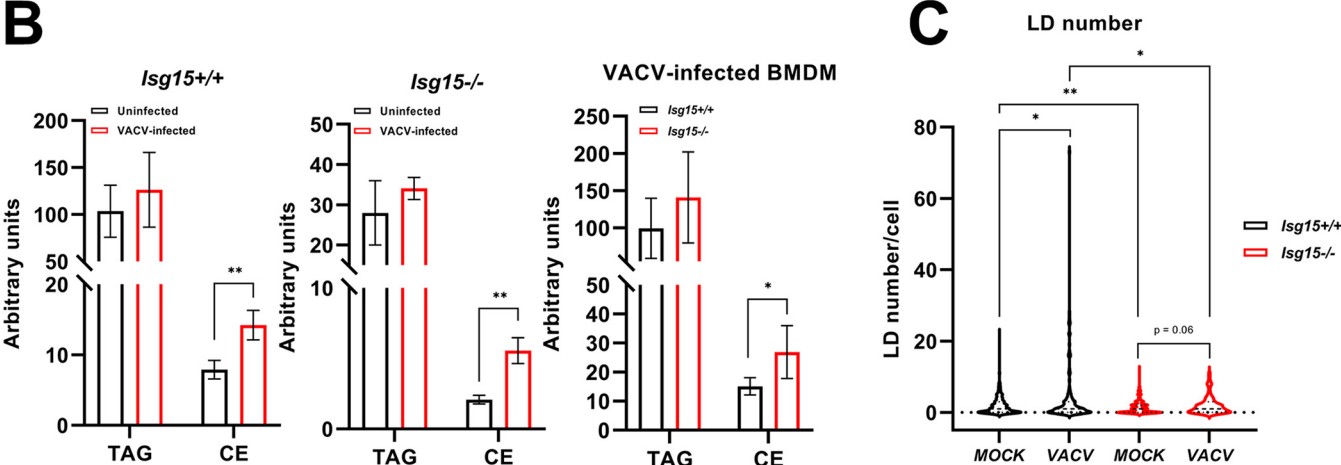

**FIG 6** Analysis of the lipid profile of VACV-infected *Isg15*$^{-/-}$ and WT BMDMs. (A and B) Lipidomic analysis of IFN-treated-, VACV-infected *Isg15*$^{-/-}$ and WT BMDMs. BMDMs from 4 mice of each genotype were subjected to UHPLC-MS-based metabolomics analysis. (A) The heatmap shows the log$_2$ fold change of the 226 metabolites analyzed for the comparisons indicated together with the *P* value obtained in the appropriate statistical analysis. Darker green and red colors indicate higher drops or elevations of the metabolites represented, respectively. Gray lines correspond to significant fold change values of individual metabolites, with darker gray colors indicating higher significances. Also, metabolites are represented in order according to their carbon number and unsaturation degree of acyl chains. (B) VACV alters the NL content of BMDMs. The levels of TAGs and CEs reported by the lipidomic analysis are shown. Mean ± SD of 4 biological replicates is represented; *, $P < 0.05$; **, $P < 0.01$; ***, $P < 0.001$. (C) VACV alters the LD content of BMDMs. IFN-treated *Isg15*$^{-/-}$ and WT BMDMs were mock infected or infected with VACV WR (1 PFU/cell for 16 h). LDs were stained with BODIPY 493/503, and cells were processed for confocal microscopy analysis. LDs were quantified with Fiji in 80 or more individual cells. Mean ± SD is represented. Student's *t* tests or Welch's *t* tests were used for the comparisons; *, $P < 0.05$; **, $P < 0.01$; ***, $P < 0.001$.

activation or inhibition of each regulator identified depending on the reported upregulation or downregulation of its downstream targets. Regarding uninfected BMDMs, we found peroxisome proliferator-activated receptor-$\alpha$ (PPAR$\alpha$), PPAR$\gamma$, PPAR$\gamma$ coactivator-1$\alpha$ (PGC-1$\alpha$), and mitochondrial transcription factor A (TFAM) among the regulators predicted to be significantly activated in *Isg15*$^{-/-}$ BMDMs (Fig. 7A). Similar results were obtained for VACV-infected WT BMDMs, with the predicted activation of regulators such as PGC-1$\alpha$ and PGC-1$\beta$ (data not shown). These transcription factors participate in the regulation of energy metabolism by stimulating mitochondrial biogenesis and catabolic processes (47), and, therefore, alterations in their expression are expected to impact cell metabolism at many different levels. Once the potential regulators responsible for the metabolic alterations observed in our cells were identified, we analyzed their expression by RT-qPCR. We selected PPAR$\alpha$ (*Ppara*), PPAR$\gamma$ (*Pparg*), PGC-1$\alpha$ (*Ppargc1a*), PGC-1$\beta$ (*Ppargc1b*), and TFAM (*Tfam*). In addition, given the marked alterations in CE levels and the relevance of cholesterol metabolism during VACV infection (32), we evaluated the expression of *Nr1h3* (encoding LXR$\alpha$), *Srebf2*, and *Scap*, key regulators of cholesterol biosynthesis.

In uninfected cells, none of the genes evaluated showed significant differences in mRNA levels between genotypes, although our data suggested increased expression of *Pparg* and *Ppargc1a* in *Isg15*$^{-/-}$ BMDMs (Fig. 7B). The higher expression of these regulators could explain the increase in mitochondrial proteins and the dysregulation of lipid metabolism observed in *Isg15*$^{-/-}$ BMDMs. However, it would be reflected by an increase in the expression of their target genes, which did not occur (Fig. S1). Hence, our data suggested that the differences in protein levels between uninfected *Isg15*$^{-/-}$ and WT BMDMs might be the result of alterations in protein homeostasis rather than variations in gene expression.

Focusing on VACV-infected BMDMs, we did not find significant differences in mRNA expression of *Ppara*, *Pparg*, *Ppargc1b*, and *Tfam* compared with uninfected BMDMs, although it should be noted that VACV infection caused a marked increase in the expression of *Pparg*, reaching higher levels in *Isg15*$^{-/-}$ BMDMs. Additionally, the expression of these genes did not significantly differ between genotypes. However, we detected a striking increase in *Ppargc1a* expression in response to infection, which was significantly higher in *Isg15*$^{-/-}$ BMDMs, which could explain the augmented levels of mitochondrial proteins and the stimulation of lipid uptake in infected macrophages and the stronger metabolic alterations in the absence of ISG15 (48). Last, attending to the expression of regulators of cholesterol homeostasis (Fig. 7B, bottom), we observed a clear decrease in *Nr1h3* and *Srebf2* mRNA levels in response to infection, consistent with a scenario where cholesterol uptake and availability are high.

Overall, our results suggest, on one hand, that the proteomic alterations observed in *Isg15*$^{-/-}$ BMDMs and VACV-infected BMDMs, although similar, are likely the result of different regulatory mechanisms; on the other hand, the lipidomic changes caused by VACV are linked to increased expression of PGC-1$\alpha$ and PPAR$\gamma$.

## DISCUSSION

Controlling the proteome essentially means controlling cell fate. This explains the great impact that the ISG15/ISGylation system has on cell functions. ISG15 participates in virtually every cellular process, and ISGylated proteins have been detected throughout the cell (20). In addition, the development of state-of-the-art techniques in protein research has recently provided an extensive list of ISGylated proteins, significantly contributing to the identification of ISG15 targets and expanding our knowledge of the cell "ISGylome" (49). Apart from ISGylation, the roles of ISG15 as a free molecule, inside and outside the cell, have substantially broadened the spectrum of action of this protein (5), opening the door to new and exciting discoveries about the ISG15/ISGylation system. Among the recent findings, the role of ISG15 as a regulator of cell metabolism stands out, both in mice and humans. In the past 6 years, several studies have demonstrated that ISG15 acts as a modulator of several pathways of cell metabolism and that

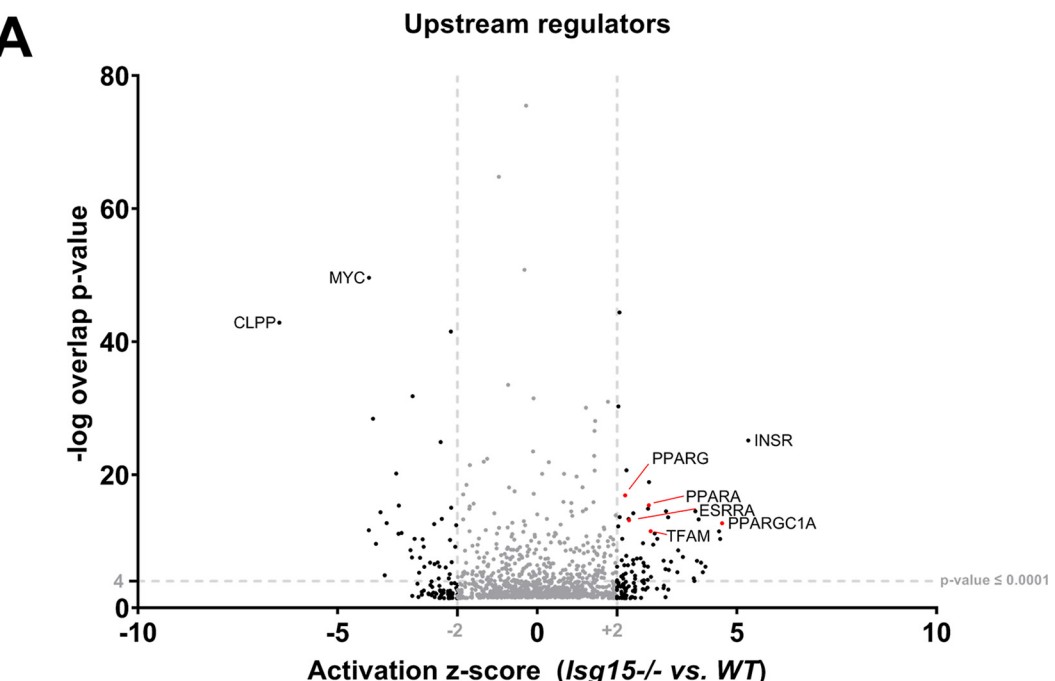

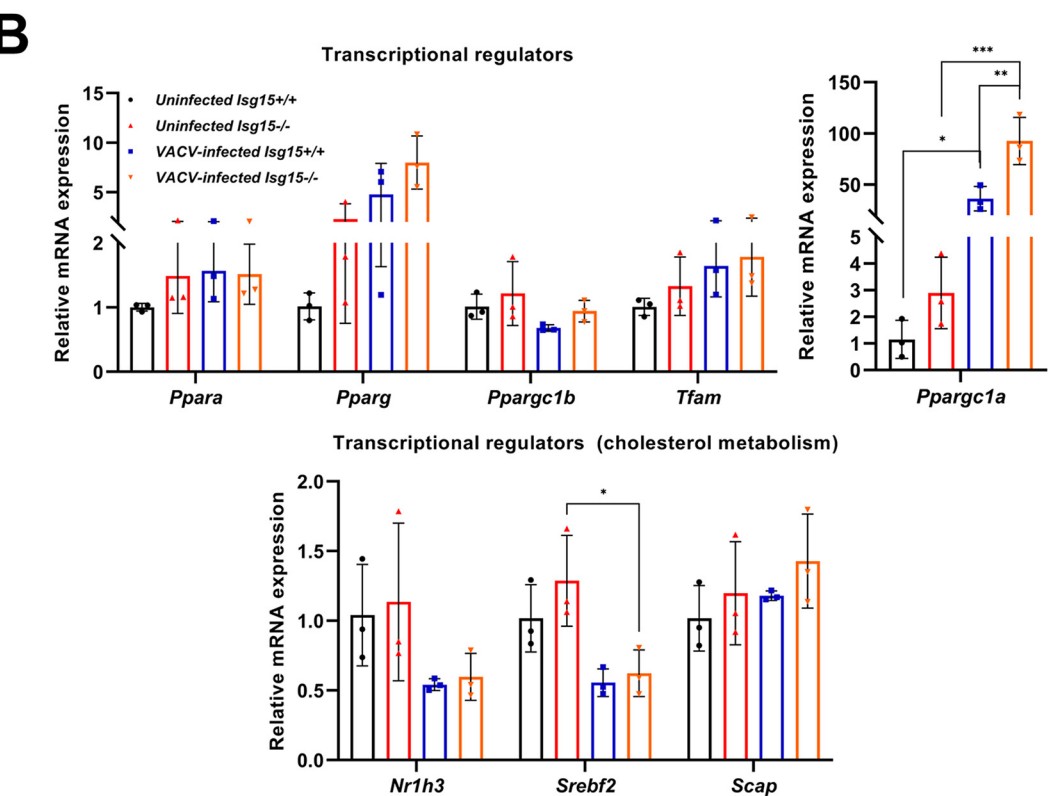

**FIG 7** Analysis of the expression of metabolic regulators in uninfected $Isg15^{-/-}$ and WT BMDMs and VACV-infected WT BMDMs. (A) $Isg15^{-/-}$ BMDMs show predicted activation of transcriptional regulators of mitochondrial biogenesis and lipid metabolism. Data from a previous proteomic analysis of IFN-treated $Isg15^{-/-}$ and WT BMDMs were subjected to IPA upstream regulators analysis. Results of the upstream regulators analysis are represented as a volcano plot, according to the activation score (x axis) and the −log (P value) of the comparison between genotypes. Relevant regulators are highlighted in red and tagged. Thresholds for activation higher and lower than +2 and −2, respectively, and a P value of <0.0001 are indicated. (B) The absence of ISG15 and VACV infection alter the expression of metabolic regulators. Total RNA was extracted from IFN-treated-, uninfected and VACV-infected BMDMs, and mRNA levels of the indicated metabolic regulators were analyzed by RT-qPCR. Expression levels were normalized to HPRT mRNA levels. Mean ± SD of 3 biological replicates is represented. Two-way ANOVA and Tukey post hoc analyses were performed for the comparisons; *, P < 0.05; **, P < 0.01; ***, P < 0.001.

its absence influences essential metabolic functions, such as OXPHOS and glycolysis (16–19, 21). These novel functions of ISG15 are closely related to its well-established antiviral activity, as viruses depend on the metabolic features of the host to complete their infectious cycle (22). For example, Kespohl and colleagues (17) showed that the ISG15-dependent reprogramming of liver metabolism conferred resistance to *Coxsackievirus* B3 infection *in vivo*. Therefore, the regulation of cell metabolism should be included in the array of indirect antiviral activities performed by ISG15, together with the modulation of protein activity and turnover, the modulation of immune responses, and the control of genome stability during viral infections (7).

In this study, we explored the role of ISG15 as a regulator of lipid metabolism in macrophages both in the presence and in the absence of VACV infection. We were motivated by our previous work in which we demonstrated that the absence of ISG15 leads to several mitochondrial alterations in BMDMs and that VACV also impacts mitochondrial functions in these cells (16). Mitochondrial activity has a direct effect on lipid metabolism, as mitochondria actively regulate lipid dynamics through the control of lipid synthesis, storage, and consumption (50, 51). In this regard, it is expected that mitochondrial defects result in alterations in lipid metabolism. Consistent with this hypothesis, our proteomic analysis revealed that the levels of many mitochondrial proteins involved in FA metabolism, as well as numerous enzymes involved in different aspects of lipid metabolism, differed between *Isg15*$^{-/-}$ and WT BMDMs. Such differences followed a clear pattern of upregulation of catabolic processes, such as FAO and lipolysis, and downregulation of anabolic processes in *Isg15*$^{-/-}$ BMDMs (Table 1). Interestingly, these changes in protein levels were not the result of increased or decreased gene expression (Fig. S1 in the supplemental material), pointing to posttranscriptional and posttranslational mechanisms as the cause of such variations in protein abundance. Both ISGylation and noncovalent interactions of ISG15 with target proteins have been shown to modulate protein stability and turnover (52, 53). Additionally, several metabolic enzymes are known to be ISGylated, including CPT1A, FASN, and FDPS (49), which are key enzymes in FAO, FAS, and cholesterol biosynthesis, respectively, and whose levels differ between *Isg15*$^{-/-}$ and WT BMDMs (Table 1). These observations suggest that ISG15 might be modulating the levels of these enzymes in BMDMs, although the effect of the ISG15/ISGylation system on these proteins is still to be determined.

Differences in the levels of lipid metabolism enzymes between genotypes were consistent with substantial changes in the whole lipid profile of *Isg15*$^{-/-}$ compared with WT BMDMs (Fig. 2A). We put our focus on NLs, as these lipid species suffered the greatest changes. NLs are nonpolar lipids that play important roles in energy storage, signaling, and maintenance of lipid homeostasis (54). NLs are stored in LDs, ubiquitous intracellular organelles consisting of an NL core surrounded by a phospholipid monolayer in association with a wide variety of proteins (55). Consistent with the reduction in NLs, *Isg15*$^{-/-}$ BMDMs showed lower LD numbers and reduced levels of PLIN2 and PLIN3 (Fig. 3; Table 1). When necessary, NLs are hydrolyzed from LDs by the action of cytoplasmic neutral lipases (lipolysis) or engulfed by autophagosomes to be degraded by lysosomal acid lipases (lipophagy). Both mechanisms generate free FAs and glycerol to be used for different purposes, predominantly the generation of ATP through FAO (56, 57). Our results pointed to increased lipolysis as the cause of reduced LD content, as suggested by increased levels of lipases in *Isg15*$^{-/-}$ BMDMs, and small or no changes in the main enzymes responsible for FA esterification were observed, whose alteration would also account for a reduction in NL content. Among lipases, the striking increase in NCEH1 in *Isg15*$^{-/-}$ BMDMs was remarkable (Fig. 3; Table 1). NCEH1 plays an important role maintaining cholesterol homeostasis in macrophages (40, 41). Several studies have demonstrated its relevance in the control of cholesterol storage, the attenuation of lipid-induced ER stress, and the restriction of atherosclerosis development through the regulation of CE levels, and, therefore, the cellular LD content (58–63). High NCEH1 levels would only explain the strong reduction in CEs observed in *Isg15*$^{-/-}$ BMDMs but not that of TAGs. However, our proteomic analysis reported an increase in lipases such as

LDAH and LIPA in $Isg15^{-/-}$ BMDMs (Table 1), which also hydrolyze TAGs and might be responsible for their reduced levels (56). Interestingly, a recent work established a relationship between ISG15 and LDs through the IFN-stimulated protein ring finger protein 213 (RNF213). RNF213 localizes to LDs, where it eliminates adipose triglyceride lipase (ATGL) from the surface, stabilizing LDs and increasing their abundance (64). Thery and coworkers demonstrated that ISGylation induced the oligomerization of RNF213 on LDs, where it recruits ISGylated proteins and exerts antimicrobial activity against bacteria and viruses (65). Surprisingly, we observed a strong reduction in RNF213 levels in the proteomics data of IFN-treated $Isg15^{-/-}$ BMDMs (data not shown), although this is not likely the cause of reduced LD content in the cells, as the authors also proposed that this phenomenon does not occur in BMDMs because $Rnf213^{-/-}$ BMDMs did not suffer variations in LD abundance compared with their WT counterparts. Therefore, these observations suggest that ISG15 might modulate the LD content by different mechanisms and in a cell-type-specific manner. Although our results pointed to increased lipolysis from LDs as the cause of reduced NL and LD content in $Isg15^{-/-}$ BMDMs, the possibility of defects in LD formation should not be ruled out. The biogenesis of LDs occurs in the ER (66) through a protein-mediated process in which seipin has a critical role (67). Additionally, proteins such as phospholipase $A_2$ isoforms have been shown to be involved in LD biogenesis at different levels, for example, by modifying the curvature of the ER membrane to facilitate LD formation (68). Interestingly, the cytosolic phospholipase $A_2$ has been shown to be ISGylated (49). Hence, the absence of ISG15 might result in altered levels or activity of these proteins, leading to defects in LD formation.

The expression of lipases is enhanced in situations of energy deprivation to release free FAs that are oxidized in mitochondria to produce ATP (56). We demonstrated that $Isg15^{-/-}$ BMDMs show reduced ATP production (Fig. 3C) (16), which might increase energy demands and, therefore, require the mobilization of stored lipids to generate ATP through FAO. However, when we measured mitochondrial respiration in the presence of etomoxir, an inhibitor of FAO, neither OCR nor ATP production were reduced in $Isg15^{-/-}$ BMDMs (Fig. 3C), indicating that these cells do not display enhanced FAO. These results were unexpected based on the increased levels of FAO enzymes reported by the proteomic analysis. If $Isg15^{-/-}$ BMDMs display high lipolytic activity, but free FAs are not metabolized, what is the destination of the free FAs and free cholesterol generated by lipolysis? Free FAs released from LDs, if not oxidized in the mitochondrion, can be reesterified and stored within LDs or can be released out of the cell to avoid lipotoxicity (57). We did not detect changes in the levels of enzymes responsible for FA esterification, meaning that reesterification could take place but likely at a slower rate than lipolysis, which would not solve the problem of accumulated free FAs and cholesterol. The transport of free FAs across the cell membrane is still a matter of debate; some authors defend a protein-mediated FA transport, whereas others argue that FAs freely diffuse across membranes following concentration gradients (69–71). The release of free cholesterol, however, is a well-studied mechanism that depends on lipid transporters, such as SR-B1, ABCA1, and ABCG1, and extracellular acceptor lipoproteins, such as APO-A1 and HDL (72). Our proteomic analysis supports the idea of enhanced cholesterol efflux in $Isg15^{-/-}$ BMDMs, as the levels of ABCA1, ABCG1, and APO-A1 are increased in these cells (Table 1). Nevertheless, further studies should be performed to elucidate the mechanisms by which the absence of ISG15 affects FA and cholesterol dynamics in macrophages.

Another unknown to solve is the cause of increased levels of lipid metabolism enzymes in $Isg15^{-/-}$ BMDMs. We showed that the mRNA levels of several lipid metabolism enzymes did not differ between genotypes (Fig. S1), which indicates posttranscriptional and/or posttranslational regulation of protein levels. One possible explanation is that the absence of ISG15 leads to defects in protein turnover. ISG15 has been shown to both dampen and stimulate protein degradation through this pathway (52, 73). Of note, it was recently demonstrated that ISG15 conjugation does not induce proteasome-mediated degradation of ISGylated proteins (74). However, ISG15 has been

shown to increase protein degradation through autophagy (10, 12, 75). We previously described alterations in autophagy in $Isg15^{-/-}$ BMDMs, which were reflected by decreased levels of several autophagy mediators, causing the accumulation of mitochondrial proteins (16). Alterations in proteostasis trigger the unfolded protein response (UPR), a stress response that integrates different signaling pathways with the common aim of restoring protein homeostasis. The UPR is usually associated with accumulation of misfolded proteins in the ER (ER stress) or in the mitochondria (mtUPR) (76, 77). Furthermore, the UPR/mtUPR can be induced by lipotoxicity due to excess free FAs and cholesterol (78), a probable situation in $Isg15^{-/-}$ BMDMs due to increased lipolysis or an excessive accumulation of NLs in the ER. In line with our hypothesis, our proteomics data suggested that $Isg15^{-/-}$ BMDMs might suffer proteostatic stress, as indicated by differential expression of proteins involved in sirtuin-, NRF2-, and eIF2$\alpha$-mediated signaling (77, 79–81) and increased levels of GRP78, a key sensor of ER stress (82). Considering these observations, it would be interesting to explore the role of ISG15 in protein dynamics and the proteostatic stress response in macrophages, which is expected to have profound implications in a wide range of cell functions.

Macrophage antiviral responses are closely linked to their metabolic features, which are reprogrammed by viruses to maximize replication. Here, we explored how VACV infection impacts BMDM metabolism and how the absence of ISG15 affects VACV-BMDM interactions from a metabolic point of view. Our proteomics data of VACV-infected WT BMDMs reported mitochondria as the main target of VACV infection, as mitochondrial dysfunction and OXPHOS were the most altered pathways in infected cells (Fig. 4). Similar observations were made by Hernáez and coworkers (83) in a transcriptomic analysis of VACV-infected L929 cells, in which genes belonging to the TCA cycle, OXPHOS, and mitochondrial dysfunction pathways showed larger differences in expression than in uninfected cells. Additionally, studies with ectromelia virus (ECTV), the etiologic agent of mousepox, showed that the expression of an ample set of mitochondrial genes was altered in both immune and nonimmune cells and that the mitochondrial network underwent profound changes during ECTV infection (84, 85). Mitochondrial metabolism is essential during VACV infection. Unlike most viruses, VACV does not require glucose to efficiently replicate but depends on glutamine metabolism to fuel the TCA cycle (29). Further studies determined that the production of palmitate through FAS, which depends on citrate from the TCA cycle, fueled mitochondrial FAO to generate ATP via OXPHOS (30). In the same line, a recent work demonstrated the importance of TCA cycle-derived citrate and FA metabolism during VACV infection and showed that stimulation of the TCA cycle is dependent on the VACV VGF (C11 protein) and activation of signal transducer and activator of transcription 3 (STAT3)-dependent signaling via epidermal growth factor receptor (EGFR)/mitogen-activated protein kinase (MAPK) and MAPK/extracellular signal-regulated kinase (ERK) signaling pathways (31, 86). We reported a striking upregulation of PGC-1$\alpha$ expression and a remarkable, yet nonstatistically significant, upregulation of PPAR$\gamma$ in infected BMDMs (Fig. 7). PGC-1$\alpha$ is a major regulator of mitochondrial biogenesis, whereas PPAR$\gamma$ is mainly involved in the regulation of lipid metabolism (47). These transcriptional regulators act downstream EGFR/MAPK and MAPK/ERK signaling pathways (87, 88), suggesting that the upregulation of PGC-1$\alpha$ in infected BMDMs may be caused by VGF expression. These observations demonstrate the close relationship between VACV and host mitochondrial and lipid metabolism. In accordance, our proteomic analysis reported that several proteins involved in FAO, lipid synthesis, lipid hydrolysis, and lipid transport and storage were differentially regulated in response to infection (Table 2). VACV provoked a scenario of increased lipid uptake and storage and decreased lipid synthesis, as indicated by higher levels of CD36, SOAT1, and PLIN2 in infected cells. Moreover, several lipases (e.g., NCEH1 and LIPA) were increased in response to infection, suggesting that VACV might stimulate lipid uptake and storage, followed by lipolysis to generate FAs to fuel FAO, which would imply a different mechanism of ATP generation than that observed in BSC40 cells and HFFs, in which FAO was fueled by FAS (30). Macrophages are efficient lipid scavengers and express many different proteins

involved in lipid uptake (89). In addition, it was demonstrated that type I IFN shifts lipid obtention mechanisms toward lipid uptake over *de novo* biosynthesis in macrophages (90), supporting this hypothesis. Increased lipid uptake and accumulation was consistent with higher NL levels and LD numbers in both infected *Isg15*$^{-/-}$ and WT BMDMs than in uninfected cells (Fig. 6). The accumulation of NLs and the activation of LD biogenesis is a common feature during infection by viral and nonviral pathogens (91). However, the effect of LD content on viral infection can be either beneficial or detrimental. On one hand, LDs have been shown to favor replication of many viruses, such as severe acute respiratory syndrome coronavirus 2 (SARS-CoV-2), rotavirus, and viruses from the *Flaviviridae* family (92–94). On the other hand, it was recently demonstrated that LD content modulates type I and III IFN signaling and that cells with reduced LD density showed decreased expression of ISGs and responded inefficiently to viral infection (95). Additionally, Bosch and coworkers (39) showed that LDs exert antimicrobial activity by direct contact with pathogens and by modulating metabolic reprogramming of infected cells to enhance immune responses. The relationship between VACV and LDs has not been explored. Considering the strong dependence of VACV on mitochondrial metabolism of lipids, it is likely that LDs are used as a source of FAs to fuel replication. In this sense, it was surprising to observe a marked reduction of CPT1A levels in VACV-infected BMDMs (Table 2) despite the increased gene expression (Fig. 5), which suggests that FAO is not enhanced during infection of BMDMs. However, several other FAO enzymes, such as ACADL, ACADVL, and HADH, showed high levels in response to infection (Table 2), which supports increased FAO. It was demonstrated that in BSC40 cells, VACV increases mitochondrial OCR due to enhanced FAO activity, as OCR increases were ablated by etomoxir treatment (30). In BMDMs, however, we demonstrated that VACV causes a reduction in OCR and ATP production (16), indicating that the metabolic reprogramming induced by VACV is cell type specific. This was demonstrated for ECTV infection, which resulted in differential expression of mitochondrial genes between L929 cells and the macrophage cell line RAW 264.7 (84).

VACV encodes a plethora of immunomodulatory genes that are expressed early and throughout infection to counteract IFN signaling (96). The mechanisms by which VACV evades the antiviral action of IFN have been recently reviewed (25) and include preventing the production of IFN (24, 97, 98), neutralizing IFN signaling through viral IFN-binding proteins (e.g., B8 and B18) (99–101), blocking the IFN intracellular signaling cascade and the production of ISGs (102, 103), and directly counteracting the action of ISGs (104, 105). Here, we showed that VACV blocks the production of ISG15 both at the mRNA and the protein levels (Fig. 5B), which was previously demonstrated to be mediated by VACV E3 protein (106). Intriguingly, VACV-infected WT BMDMs and uninfected *Isg15*$^{-/-}$ BMDMs share diverse phenotypic features when subjected to IFN treatment, as disclosed by the comparison of the proteomic analyses of VACV-infected WT BMDMs (Fig. 4) and uninfected *Isg15*$^{-/-}$ BMDMs and the results of our previous research (16). Both present increased levels of mitochondrial proteins, impaired OXPHOS and ROS production, and a heterogeneous activation phenotype (16). Based on these observations, we hypothesize that the absence of ISG15, or reduced ISG15/ISGylation due to VACV infection, dampens the ability of BMDMs to modulate the IFN-induced reprogramming of mitochondrial metabolism, leading to mitochondrial dysfunction. As well, focusing on lipid metabolism, we propose that the absence of ISG15 determines the magnitude of the alterations in lipid metabolism caused by VACV, given that *Isg15*$^{-/-}$ BMDMs showed enhanced expression of the metabolic regulators PGC-1$\alpha$ and PPAR$\gamma$ (Fig. 7) and greater changes in the lipid profile in response to VACV infection. These results suggest that ISG15 is an important host factor to restrain the effects of VACV on macrophage metabolism and highlight the relevance of the ISG15-mediated modulation of metabolism to efficiently counteract viral infections.

## MATERIALS AND METHODS

**Cell culture.** BSC40 cells (African green monkey kidney cells, ATCC, CRL-2761) and primary WT and *Isg15*$^{-/-}$ mouse bone marrow-derived macrophages (BMDMs) were used. Cells were grown in 1× Dulbecco's modified Eagle's medium (DMEM) supplemented with fetal bovine serum (FBS; 5% for BSC40

cells and 10% for BMDMs), 2 $\mu$g/mL glutamine (Sigma, G7513), 1× nonessential amino acids (Sigma, M7145), 1× penicillin-streptomycin (Gibco, 15140-122), and 1 $\mu$g/mL amphotericin B (Gibco, 15290-018). Cells were incubated in a humidified incubator at 37°C and 5% $CO_2$.

**Isolation and culture of BMDMs.** Eight- to 12-week-old female mice from the C57BL/6J (WT) and B6.129P2-Isg15tm1Kpk/J ($Isg15^{-/-}$) strains were used to isolate BMDMs. The protocol followed for BMDM isolation is based on a well-established method for mouse bone marrow extraction (107, 108). Mice were euthanized following the appropriate euthanizing procedure and soaked with 70% ethanol before dissection. The tibia and femur bones were obtained by dislocation of their respective joints. When possible, the ilium (pelvic bone) was also extracted. The muscles that remained attached to the bones were removed by scraping with a scalpel. Once the bones were completely clean, the epiphyses were cut off, and the bones were placed in a 0.5-mL centrifuge tube with a hole at the bottom. The tubes containing the bones were placed into 1.5-mL centrifuge tubes, and the bone marrow was extracted by centrifugation at 2,000 × $g$ for 10 s. Bone marrow pellets were resuspended in 1 mL of BMDM culture medium consisting of 1× DMEM containing 10% FBS and 10 ng/mL recombinant murine macrophage colony-stimulating factor (rM-CSF; PeproTech, 315-02), and the cell suspension was transferred to a 50-mL tube containing the appropriate amount of BMDM culture medium. Cells were seeded on 100-mm cell culture dishes or 6-well, 12-well, or 24-well tissue culture plates depending on the requirements of the experiment, following the ratio of 4 plates per mouse. Plates were then placed in a humidified incubator at 37°C and 5% $CO_2$ and incubated for 6 days. After 6 days of differentiation, BMDMs attached to the surface of the plate were washed with phosphate-buffered saline (PBS), and the adequate volume of 1× DMEM containing 10% FBS was added. BMDMs were then subjected to the specific treatments and procedures of each experiment (specified elsewhere).

**Generation and purification of a viral stock.** BSC-40 cells were propagated in 150-mm tissue culture dishes and scaled up to 20 to 25 dishes, following the specified culture conditions. When 80% confluency was reached, cells were infected with VACV wild-type Western Reserve strain (VACV WR) at 0.01 PFU/cell for 48 h. At 48 h postinfection (hpi), infected cells were harvested with a cell scraper, and the cell suspension was transferred to sterile 50-mL tubes. Cells were pelleted by centrifugation at 250 × $g$ and 4°C for 10 min, and the supernatant was discarded. Pellets were resuspended in 10 mL of 10 mM Tris-HCl (pH 9.0), and cell suspensions were sonicated in a water bath using a Branson Ultrasonics sonifier 450; sonication was continuous at 50% amplitude until cell clumps were completely homogenized. After sonication, the homogenized cell suspension was transferred to 15-mL tubes and centrifuged for 5 min at 400 × $g$ and 4°C. Virus-containing supernatants were transferred to new 50-mL tubes, and cell pellets were resuspended in 10 mL of 10 mM Tris-HCl (pH 9.0), sonicated again under the same conditions described above, and centrifuged for 5 min at 400 × $g$ and 4°C. Virus-containing supernatants were mixed with the supernatants previously obtained. A sucrose cushion was prepared by adding 20 mL of cold 45% sucrose in 10 mM Tris-HCl (pH 9.0) in an ultracentrifuge tube (Ultra-Clear centrifuge tubes, Beckman Coulter, 344058). Virus-containing supernatant was slowly dispensed on top of the sucrose cushion, and ultracentrifuge tubes were placed into the rotor buckets and equilibrated with the appropriate volume of 10 mM Tris-HCl (pH 9.0) before ultracentrifugation. Supernatants were ultracentrifuged for 1 h at 72,100 × $g$ and 4°C using a Beckman SW28 Ti swinging-bucket aluminum rotor. After ultracentrifugation, supernatants were carefully discarded, and virus pellets were resuspended in 5 mL of 10 mM Tris-HCl (pH 9.0) and frozen at −80°C. To obtain a highly purified virus stock, a further purification step was performed. A sucrose gradient was prepared by adding 16 mL of 45% sucrose in 10 mM Tris-HCl (pH 9.0) in an ultracentrifuge tube and carefully dispensing 16 mL of 20% sucrose in 10 mM Tris-HCl (pH 9.0) on top. The sucrose mixture was frozen at −80°C for at least 5 h and thawed overnight at 4°C, causing the formation of a sucrose gradient. Virus suspension was thawed on ice, carefully pipetted over the sucrose gradient, and ultracentrifuged for 20 min at 40,000 × $g$ and 4°C using a Beckman SW28 Ti swinging-bucket aluminum rotor. After ultracentrifugation, the virus-containing white band in the middle of the sucrose gradient was recovered by aspiration with a glass pipette and dissolved in 10 mM Tris-HCl (pH 9.0). Virus suspension was ultracentrifuged for 45 min at 72,100 × $g$ and 4°C using a Beckman SW28 Ti swinging-bucket aluminum rotor. Finally, the supernatant was discarded, the virus pellet was resuspended in 1 mL of 10 mM Tris-HCl (pH 9.0), and 100-$\mu$L aliquots were prepared and stored at −80°C.

**Viral infection.** Frozen viral stocks were placed on ice until completely thawed. Before infection, viral stocks were sonicated in a water bath to dismantle virus aggregates by using a Branson Ultrasonics sonifier 450. The sonication protocol consisted of three cycles of 10 s of continuous sonication followed by 10 pulses at 50% amplitude. Next, viral dilutions were prepared in FBS-free 1× DMEM to match the desired MOI. Cells were washed with PBS and infected with the appropriate inoculum volume, which was 3 mL, 1.5 mL, 600 $\mu$L, 300 $\mu$L, and 150 $\mu$L for 150-mm and 100-mm tissue culture dishes and 6-well, 12-well, and 24-well tissue culture plates, respectively. Infected plates were incubated for 1 h to promote virion adsorption to the cell surface and gently shaken every 15 min to distribute the inoculum. After the adsorption time, the inoculum was removed, cells were washed with PBS, and the infection was restricted with the appropriate volume of 1× DMEM containing 2% FBS. Infected plates were incubated for the desired infection times. At the convenient postinfection times, cells were processed following the requirements of each experiment.

**Interferon treatment.** Cells were washed with PBS and treated for 16 or 24 h with the appropriate volume of 1× DMEM containing 10% FBS and 500 U/mL of universal type I interferon (PBL Assay Science, 11200-2) at 37°C and 5% $CO_2$ in a humidified incubator.

**Immunoblotting.** Cells were homogenized in lysis buffer (50 mM Tris, 150 mM NaCl, and 1% NP-40) supplemented with protease inhibitor (cOmplete mini, Roche, 11836153001) and phosphatase inhibitor

(Pierce phosphatase inhibitor mini tablets, Thermo Scientific, A32957) cocktails. To ensure a complete homogenization and protein extraction, samples underwent three cycles of freezing in dry ice and thawing at 37°C, and resulting cell debris was pelleted by centrifugation at 21,000 × $g$ for 20 s. Protein-containing supernatants were transferred to new centrifuge tubes, and protein concentration was determined by Bradford assay using protein assay dye reagent concentrate (Bio-Rad, 500-0006) and measuring absorbance at a wavelength of 595 nm. Protein samples were mixed with SDS sample buffer (Laemmli buffer) and boiled at 95°C for 5 min. Proteins were separated by 12% or 10% SDS-PAGE at an intensity of 40 mA per gel using hand-cast gels and Tris-glycine running buffer (25 mM Tris, 152 mM glycine, and 3.5 mM SDS [pH 8.5]). Protein marker VI (10-245) prestained (PanReac AppliChem, ITW Reagents, A8889) was used as marker for protein molecular weight. Proteins in the gel were transferred to a 0.45-$\mu$m polyvinylidene difluoride (PVDF) membrane (Merck Millipore, IPVH00010) in a semidry immunoblot transfer system (Trans-Blot SD semidry transfer cell, Bio-Rad, 1703940) at 15 V for 45 min. After transfer, membranes were blocked with 5% skimmed milk in Tris-buffered saline (TBS; pH 7.5) with 0.1% Tween 20 (TBS-T) for 1 h at room temperature (RT) with agitation. Blocked membranes were incubated with the desired primary antibodies overnight at 4°C with agitation. Incubation with primary antibodies was followed by three 5-min washing steps with TBS-T and incubation with the appropriate horseradish peroxidase (HRP)-conjugated secondary antibodies for 1 h at RT with agitation. Primary and secondary antibodies (Table S1 in the supplemental material) were prepared in 0.5% skimmed milk in TBS-T, and the concentrations are specified in figure legends. Finally, membranes were washed three times in TBS-T (5 min each) and developed by incubating with chemiluminescence substrate (Clarity Western ECL substrate, Bio-Rad, 1705061) for 1 min with agitation in the dark. Imaging was performed using a ChemiDoc MP imaging system (Bio-Rad, 1708280) and Image Lab software (Bio-Rad, 1709690).

**Analysis of mRNA expression by two-step quantitative reverse transcription-PCR (RT-qPCR).** Total RNA was extracted using NucleoZOL (Macherey-Nagel, 740404.200), following the manufacturer's instructions. RNA pellets were resuspended in diethyl pyrocarbonate (DEPC)-treated RNase-free water (Thermo Fisher Scientific, J70783.K2), and RNA concentration was measured using a Maestrogen MaestroNano Spectrophotometer (MaestroGen). One microgram of RNA was reverse transcribed using the high-capacity cDNA reverse transcription kit (Applied Biosystems, 4368814). For mRNA quantification, reactions were prepared with 10 $\mu$L of 2× qPCRBIO SyGreen mix Hi-ROX (PCR Biosystems, PB20.12-05), 400 nM forward and reverse primers (Table S2), 25 ng of template cDNA, and the adequate volume of DEPC-treated RNase-free water up to a final volume of 20 $\mu$L. Analysis was performed in a StepOnePlus real-time PCR system (Thermo Fisher Scientific, 4376600) using the following protocol: preincubation for 2 min at 95°C (1 cycle), denaturation at 95°C for 5 s, and annealing and extension at 60°C for 30 s (40 cycles). Melting curve analysis was performed by melting at 95°C for 15 s, 65°C for 60 s, and 95°C for 15 s and stepping and holding (1 cycle), followed by a step of cooling at 4°C for 30 s. The *HPRT* gene was used as a housekeeping gene (Table S2), and the relative gene expression was calculated by the threshold cycle ($2^{-\Delta\Delta CT}$) method.

**Analysis of mitochondrial respiratory parameters.** Real-time oxygen consumption rate (OCR) and extracellular acidification rate (ECAR) were assessed for WT and *Isg15*$^{-/-}$ BMDMs using a Seahorse XF-96 extracellular flux analyzer platform (Agilent); 6 ×$10^4$ BMDMs/well in five wells were used per condition. Measurements were performed in XF base medium (Agilent) supplemented with 5 mM glucose, 2 mM glutamine, 1 mM sodium pyruvate, 100 $\mu$g/mL penicillin, and 100 $\mu$g/mL streptomycin. Where indicated, cells were treated with type I IFN (500 U/mL) 16 h before the analysis. In addition, to assess specific metabolite demands, cells were treated with etomoxir (5 $\mu$M final concentration) for 30 min before the analysis started. The electron transport chain (ETC) inhibitors oligomycin (oligo; 2 $\mu$M final), carbonyl cyanide-*p*-trifluoromethoxyphenylhydrazone (FCCP; 2.85 $\mu$M final), rotenone (Rot; 1 $\mu$M final), and antimycin A (AA; 2 $\mu$M final) were used to assess the different mitochondrial respiratory parameters. Basal respiration was defined as OCR in the absence of inhibitors. Maximal respiration was defined as the OCR after addition of oligomycin and FCCP. The protocol consisted of five measurements under basal conditions, oligo injection, three measurements, first FCCP injection, three measurements, second FCCP injection, three measurements, injection of Rot + AA, three measurements. Measurements lasted 3 min and were preceded by 2 min of mixing. Once finished, cells were fixed with 4% paraformaldehyde (PFA) for 10 min at RT, washed with PBS, and stained with 0.5 ng/mL Hoechst stain solution (Sigma, H6024) for 15 min at RT in the dark. Cell counts were determined by fluorescence microscopy using an Operetta CLS high-content analysis system (PerkinElmer), and results were normalized to cell numbers. Data were analyzed using the Wave software (Agilent).

**Lipidomic analysis of BMDMs. (i) Sample obtention.** BMDMs from WT and *Isg15*$^{-/-}$ mice (4 mice per genotype) were isolated and cultured as described above. After 6 days, cells were treated with type I IFN (500 U/mL) for 24 h or infected with VACV WR at 1 PFU/cell for 16 h. BMDMs were then collected and pelleted by centrifugation at 250 × $g$ for 10 min, and pellets were frozen at −80°C until they were processed for lipidomic analysis at the Center for Cooperative Research in Biosciences (CIC bioGUNE, Derio, Basque Country, Spain).

**(ii) Metabolite extraction and sample preparation.** Metabolite extraction was accomplished by fractionating the BMDMs into pools of species with similar physicochemical properties, using appropriate combinations of organic solvents. Cell pellets were resuspended in cold water and vortexed. Proteins were precipitated from the lysed cell samples by adding methanol, followed by addition of chloroform after a brief vortex. Both extraction solvents were spiked with metabolites not detected in unspiked cell extracts (internal standards). Samples were incubated at −20°C for 30 min, and, after vortexing, 500 $\mu$L was collected for liquid chromatography-tandem mass spectrometry (LC-MS/MS) analysis. Cell extracts were mixed with ammonium hydroxide in water (pH 9.0), vortexed, and incubated for 1 h

at $-20°C$. Samples were centrifuged at 18,000 $\times$ $g$ and 4°C for 15 min, and the organic phase was collected and dried under vacuum. Dried extracts were reconstituted in acetonitrile:isopropanol (1:1), resuspended by agitation for 10 min, centrifuged at 18,000 $\times$ $g$ and 4°C for 5 min, and transferred to plates for ultrahigh-performance liquid chromatography-mass spectrometry (UHPLC-MS) analysis (109). Additionally, two different types of quality control (QC) samples were used to assess the data quality (110). The QC samples are reference serum samples, which were evenly distributed over the batches and extracted and analyzed together with the individual samples. A QC calibration sample was used to correct the different response factors between and within batches, and a QC validation sample was used to assess how well data preprocessing procedures improved data quality.

**(iii) UHPLC-MS analysis.** For this analytical platform, randomized sample injections were performed, with each of the QC samples uniformly interspersed throughout the batch run. The overall quality of the procedure was monitored using repeat extracts of the QC samples. Retention time stability throughout the run was generally less than 6 s variation (injection-to-injection), and mass accuracy was generally less than 5 ppm for mass-to-charge ratio ($m/z$) of 400 1,000 and less than 1.2 mDa for $m/z$ 50 to 400.

**(iv) Data preprocessing, *normalization, and quality control.*** All data were processed using the TargetLynx Application Manager for MassLynxTM 4.1 software (Waters Corp., Milford, USA). A set of predefined retention time-$m/z$ pairs (Rt-$m/z$) corresponding to metabolites included in the analysis was fed to the program. Associated extracted ion chromatograms (mass tolerance window = 0.05 Da) were then peak detected and noise reduced in both the LC and MS domains so that only true metabolite-related features were processed by the software. A list of chromatographic peak areas was generated for each sample injection. Normalization factors were calculated for each metabolite by dividing their intensities in each sample by the recorded intensity of and appropriate internal standard in the same sample, following the procedure described in reference 111. The most appropriate internal standard for each variable was defined as that which resulted in a minimum relative standard deviation after correction, as calculated from the QC calibration samples over the analysis batches. In addition, robust linear regression was used to estimate any intrabatch drift not corrected by internal standard correction in the QC calibration samples. Following normalization, response values were assessed. Where coefficients of variation higher than 30% were found, corresponding sample injection data automatically generated by the software were manually revised, and modifications were performed where appropriate. Any variables with zero values in the corrected data set were replaced with missing values before forming the final data set and performing the statistical analyses.

**(v) Statistical analysis.** Once normalized, the dimensionality of the complex data set was reduced to enable easy visualization of any metabolic clustering of the different groups of samples. This was achieved by multivariate data analysis, including nonsupervised principal-component analysis (PCA) and/or supervised orthogonal partial least-squares to latent structures (OPLS) approaches (112). To examine potential metabolic differences between infected and uninfected samples of the same genotype and between $Isg15^{-/-}$ and WT uninfected samples, we performed univariate statistical analyses. Group percentage changes and paired Student's $t$ test $P$ values were calculated, and a Welch's $t$ test was performed where unequal variances were found. When comparing $Isg15^{-/-}$ versus WT infected samples, and to avoid the differences per mouse, data were first normalized by dividing the data of each metabolite obtained for infected samples per the data obtained for uninfected samples for the same mouse. Then, univariate statistical analyses (Student's $t$ test $P$ value and Welch's $t$ test $P$ value) were performed.

**OA-FBS conjugation.** Oleic acid (OA; Sigma-Aldrich, O1383) was dissolved in ethanol to a final concentration of 400 mM. Aliquots of FBS were prepared and placed in a water bath at 57°C. OA was added to the FBS aliquots to constitute a 2 mM OA solution. A total of 10 OA additions were done while vortexing; the first 6 additions were followed by 20-min incubations at 57°C in a water bath, increasing the incubation time to 30 min in the last 4 additions. Vehicle control solutions were prepared by adding the appropriate volume of ethanol to FBS. Once prepared, solutions were aliquoted and stored at $-80°C$.

**LD analysis by confocal microscopy.** BMDMs were seeded on coverslips in 12-well or 24-well plates at a density of 2 $\times10^5$ or 1 $\times10^5$ cells per well, respectively. Cells were washed with PBS and treated or not with type I IFN in 1$\times$ DMEM containing 10% FBS. LD synthesis was induced by incubating cells with 1$\times$ DMEM containing 10% FBS and 100 $\mu$M OA together with type I IFN treatment. Cells were incubated for 24 h in a humidified incubator at 37°C and 5% $CO_2$. For LD staining, BMDMs were washed twice with PBS and incubated with 1$\times$ DMEM containing 10% FBS and BODIPY 493/503 to a final concentration of 1 $\mu$g/mL for 30 min in a humidified incubator at 37°C and 5% $CO_2$. After staining, cells were washed twice with PBS and fixed with 300 $\mu$L of fixation buffer (4% paraformaldehyde [PFA] in PBS) for 20 min at RT. After fixation, PFA was aspirated, and cells were washed five times with PBS, incubated with 1 $\mu$g/mL 2-(4-amidinophenyl)-1$H$-indole-6-carboxamidine (DAPI) for 20 min at RT in the dark, and washed another five times with PBS. Finally, coverslips were mounted on glass slides using ProLong Diamond mounting medium (Invitrogen, Thermo Fisher Scientific, P36961), incubated overnight at RT in the dark, and stored at 4°C until analysis. Confocal microscopy analyses were performed using Leica TCS SP8 and Zeiss LSM 880 Airyscan superresolution confocal microscopes. Processing of microscopy images was performed with Fiji software (75), and Fiji and Aivia AI image analysis software (https://www.aivia-software.com) were used for image analysis.

**Analysis of lipid mobilization by confocal microscopy.** WT and $Isg15^{-/-}$ BMDMs were seeded on 8-well chamber slides ($\mu$-Slide 8-well, ibidi, 80826) at a density of 1.5 $\times10^5$ cells per well. The culture medium was replaced with 1$\times$ DMEM containing 10% FBS and 1 $\mu$M BODIPY 558/568 C12, and cells were incubated for 16 h at 37°C and 5% $CO_2$. After incubation, cells were washed three times with growth medium, and fresh medium (1$\times$ DMEM containing 2% FBS) was added. Cells were subjected to confocal

microscopy analysis with live cells with a Leica TCS SP8 confocal microscope. Images were taken at 0 and 3 h after medium was replaced. Microscopy images were processed and analyzed with Fiji.

**Proteomic analysis of BMDMs.** Protein extracts from WT and *Isg15*$^{-/-}$ BMDMs, or VACV-infected WT BMDMs, treated with type I IFN (500 U/mL, 24 h) were obtained after cell lysis in extraction buffer (50 mM Tris-HCl, 1 mM EDTA, and 1.5% SDS [pH 8.5]). Samples digested in trypsin and the resulting peptides were subjected to 4-plex isobaric labeling (iTRAQ) and were separated into 8 fractions by cation exchange chromatography using Waters Oasis MCX cartridges (Waters Corp., Milford, MA, USA) and graded concentrations of ammonium formate (pH 3.0; AF3) in acetonitrile (ACN). The tryptic peptide fractions were subjected to nanoLC-MS/MS. High-resolution analysis was performed on a nano-HPLC Easy nLC 1000 liquid chromatograph (Thermo Scientific, San Jose, CA, USA) coupled to an Orbi-trap Fusion mass spectrometer (Thermo Scientific). Protein identification was performed using the SEQUEST HT algorithm integrated in Proteome Discoverer 1.4 software (Thermo Scientific). MS/MS scans were matched against a mouse database (UniProtKB/Swiss-Prot 2015_11 release). Peptides were identified from MS/MS data using the probability ratio method (113). False discovery rate of peptide identifications was calculated by the refine method (114). Quantitative information was extracted from the MS/MS spectra of iTRAQ-labeled peptides. For comparative analysis of protein abundance changes, the weighted scan-peptide protein (WSPP) statistical workflow was applied. The quantified proteins were functionally annotated using the Ingenuity Knowledge Database (115) and DAVID, a repository that includes 13 functional databases, including Panther, KEGG, and Gene Ontology.

The proteins identified in the proteomic analysis of BMDMs and the ratio fold changes between *Isg15*$^{-/-}$ and WT cells and between VACV-infected and uninfected WT cells were used to analyze the differences in the proteome related to the absence of ISG15 or VACV infection. Ingenuity Pathway Analysis software (IPA; Qiagen) was used to explore the main pathways and regulators affected by the lack of ISG15 and VACV infection. The canonical pathway analysis identified the pathways from the IPA library that were most significant to our data set. The statistical significance of the association between identified pathways and our data was determined by Fisher's exact test, for which the significance threshold was set at 0.05 (116). The upstream regulator analysis identifies proteins, present or not in the data set, that are potential master regulators of the different pathways represented in the data set. This tool sets an overlap *P* value to evaluate the enrichment of proteins in the data set related to a given master regulator from the database.

**Statistical analysis.** Statistical analyses were performed with GraphPad Prism V 9.0 software (GraphPad Software, San Diego, CA, USA; http://www.graphpad.com). Comparisons of two groups were analyzed applying the two-tailed unpaired Student's *t* test or the Welch's *t* test when variances were unequal. Equality of variances was tested with the *F* test. When necessary, two-way analysis of variance (ANOVA) and Tukey *post hoc* tests were performed. A *P* value of <0.05 was considered statistically significant.

## SUPPLEMENTAL MATERIAL

Supplemental material is available online only.

**SUPPLEMENTAL FILE 1**, PDF file, 0.5 MB.

## ACKNOWLEDGMENTS

We thank the expert technical assistance of Sara Sandoval. We are grateful to Miguel Sánchez-Álvarez who has kindly provided several commercial reagents. We would like to thank the Spanish National Plan for Scientific and Technical Research and Innovation (Plan Estatal de Investigación Científica y Técnica y de Innovación), (Ministry of Health of Spain, State Secretary of R+D and FEDER/FSE).

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
