## [Reviewer comments · Microbiology Spectrum]

Microbiology Spectrum

ISG15 is a novel regulator of lipid metabolism during *Vaccinia virus* infection

Manuel Albert, Jesús Vázquez, Juan Falcón-Pérez, María A. Balboa, Marc Liesa, Jesús Balsinde, and Susana Guerra

Corresponding Author(s): Susana Guerra, Universidad Autónoma

Review Timeline:

Submission Date:	September 23, 2022
Editorial Decision:	November 2, 2022
Revision Received:	November 8, 2022
Accepted:	November 10, 2022

Editor: Miguel Martinez

Reviewer(s): Disclosure of reviewer identity is with reference to reviewer comments included in decision letter(s). The following individuals involved in review of your submission have agreed to reveal their identity: Huimin Hui (Reviewer #2)

Transaction Report:

DOI: <https://doi.org/10.1128/spectrum.03893-22>

November 2, 2022

Dr. Susana Guerra
Universidad Autónoma
Departament of Preventive Medicine and Public and Microbiology
Arzobispo Morcillo s/n
Madrid, Madrid 28049
Spain

Re: Spectrum03893-22 (ISG15 is a novel regulator of lipid metabolism during viral infection)

Dear Dr. Susana Guerra:

Your manuscript has been considered by two reviewers recruited for their expertise in the field. As their comments indicate, these individuals felt that your manuscript contained interesting observations but that it required modification before it could be considered acceptable for publication. In addition, reviewers suggested to send to Public repository details the following data: 1) Metabolomic analysis of WT and ISG15-/- BMDMs, 2) Lipid profile of Vacv infected WT and ISG15-/- BMDMs and 3) the data source of tables 1 and 2.

Link Not Available

Sincerely,

Miguel Martinez

Journals Department
Reviewer comments:

Reviewer #1 (Comments for the Author):

The study by Albert et al, explains very well the role of ISG15 in viral infection. By performing a detailed Proteomic analysis the authors have confirmed alteration in signaling pathways responsible for lipid metabolism in ISG15-/- BMDMs, Later they have demonstrated it under infection with vaccinia virus.

There are some questions which needs to be addressed for to make the study concrete:

1. The authors should explain why only vaccinia virus was used for the study ? Whereas they could have tested with other virus types such as non-enveloped viruses to demonstrate it as a general phenomena.

2.

2.

Reviewer #2 (Comments for the Author):

This paper reports some interesting findings showing that ISG15 is a novel regulator of lipid metabolism during viral infection. The study is well-designed and data is overall solid. However, there are several weaknesses may help improve the study.

1.Tables 1 and 2 in this study chose IFN-treated ISG15^{-/-} and WT BMDM as control groups, why not untreated BMDM? No details are provided. As well, the proteomic data usually included a large number of peptides, how the proteins from tables 1 and 2 were obtained?

2.The authors suggest ISG15 modulates the levels of proteins involved in lipid metabolism independently of gene expression, however, the several genes selected are not contained both in IFN-treated and VACV-infected samples. Is this conclusion accuracy?

3.Figure 4 only compared the proteomic profile of VACV-infected WT and uninfected ISG15^{-/-}? What happened to other groups?

Staff Comments:

Preparing Revision Guidelines

Please return the manuscript within 60 days; if you cannot complete the modification within this time period, please contact me. If you do not wish to modify the manuscript and prefer to submit it to another journal, please notify me of your decision immediately so that the manuscript may be formally withdrawn from consideration by Microbiology Spectrum.

Madrid, November 2th, 2022

Miguel Martínez
Editor, Microbiology Spectrum
Journals Department
Dear Dr., Martínez,

Please find enclosed our revised manuscript Spectrum03893-22 entitled "**ISG15 is a novel regulator of lipid metabolism during viral infection**", which addresses all the recommendations and criticisms made by the reviewers. We would like to thank the reviewers for their critical comments and suggestions, which have helped to improve the manuscript significantly. Below, we detail a point-by-point response to these open questions:

Reviewer #1 (Comments for the Author):

The study by Albert et al, explains very well the role of ISG15 in viral infection. By performing a detailed Proteomic analysis, the authors have confirmed alteration in signaling pathways responsible for lipid metabolism in ISG15^{-/-} BMDMs, Later they have demonstrated it under infection with vaccinia virus.

There are some questions which needs to be addressed for to make the study concrete:

1. The authors should explain why only vaccinia virus was used for the study? Whereas they could have tested with other virus types such as non-enveloped viruses to demonstrate it as a general phenomena.

Answer: Following the reviewer suggestion, we have now indicated that we focused our studies on VACV infection as a continuation of previous work where we demonstrated the role of ISG15 in mitochondrial functionality in VACV-infected bone marrow-derived macrophages (BMDM) (**lines 98-100**).

The reviewer is totally right indicating that we have not demonstrated the role of ISG15 regulating lipid metabolism during infection with other types of viruses. For this reason, we have changed the title of the revised manuscript and we have now indicated that the work has been focused on VACV infection.

Reviewer #2 (Comments for the Author):

This paper reports some interesting findings showing that ISG15 is a novel regulator of lipid metabolism during viral infection. The study is well-designed and data is overall solid. However, there are several weaknesses may help improve the study.

1. Tables 1 and 2 in this study chose IFN-treated ISG15^{-/-} and WT BMDM as control groups, why not untreated BMDM? No details are provided. As well, the proteomic data usually included a large number of peptides, how the proteins from tables 1 and 2 were obtained?

Answer: Following the reviewer suggestion, we have now indicated that the treatment of IFN stimulates the expression of ISG15 and favors the ISGylation (**lines 108-110**). ISG15 and ISGylation are found at greatly reduced levels in unstimulated conditions (PMID: 29077752). In

Baldanta et al., 2017 we evaluated in uninfected BMDM ISGylation in total, or cytoplasmic or mitochondrial proteins extracts, and we clearly observed ISGylated proteins in all fractions and their levels increased following IFN pre-treatment, but not in untreated cells. This prompted us to stimulate BMDM with type I IFN, so we could enhance the effects of ISG15/ISGylation on the proteome.

As the reviewer suggested, in the revised version we have provided further details regarding the proteomic analysis of BMDM, including protein annotation and identification methods that helped in the elaboration of Tables 1 and 2 (**Results – lines 110-115 and 250-255; Materials and Methods – lines 843-867**).

2. The authors suggest ISG15 modulates the levels of proteins involved in lipid metabolism independently of gene expression, however, the several genes selected are not contained both in IFN-treated and VACV-infected samples. Is this conclusion accuracy?

Answer: As the reviewer points out, Tables 1 and 2 do not represent the same genes, as there are some of them that only appear in uninfected BMDM or in VACV-infected BMDM. These differences between tables are due to our decision to represent only those genes whose fold change (FC) values were higher than 2 and lower than -2. For example, the FC value of *Soat1* in VACV-infected WT cells is 11.56 (Table 2), whereas in uninfected *Isg15*^{-/-} cells it is 0.42 (see Supplementary material – BMDM Proteomics). Therefore, we decided not to include this protein in Table 1. However, we considered including this and other genes in the analysis due to their relevance in lipid metabolism.

3. Figure 4 only compared the proteomic profile of VACV-infected WT and uninfected ISG15^{-/-}? What happened to other groups?

Answer: Figure 4 compares the proteomic profiles of VACV-infected and uninfected **WT** BMDM. We are aware of our lack of information about other conditions, such as VACV-infected *Isg15*^{-/-} BMDM, which would have provided useful data about the effects of both VACV infection and the absence of ISG15, what would have enriched our study. However, our resources are very limited, and we decided to focus our efforts and expenses on those proteomics experiments that are essential for the development of our objectives. Moreover, as it was demonstrated that VACV modulates the ISG15/ISGylation system (PMC3911545), we thought it would be interesting to compare the effects of the absence of ISG15 and VACV infection on the macrophage proteome. This made us choose the conditions presented in this work. However, we explored the effects of these conditions in the expression of some genes involved in lipid metabolism by RT-qPCR, what helped in the interpretation of our results.

We would like to thank the reviewers for their critical comments and suggestions, which have helped to significantly improve the manuscript. I believe that with these corrections and modifications incorporated, the revised manuscript addresses the criticism of the reviewers. I hope that you now will consider the paper suitable for publication in the Microbiology Spectrum.

Yours sincerely,

Susana Guerra /Assistant Professor in Microbiology
Department of Preventive Medicine, Public Health and Microbiology
Laboratory D-9, School of Medicine
Autonoma University of Madrid
St Arzobispo Morcillo, sn 28029
Madrid (SPAIN)
Phone: +34 91 497 5440

Staff Comments:

Preparing Revision Guidelines

Please return the manuscript within 60 days; if you cannot complete the modification within this time period, please contact me. If you do not wish to modify the manuscript and prefer to submit it to another journal, please notify me of your decision immediately so that the manuscript may be formally withdrawn from consideration by Microbiology Spectrum.

November 10, 2022

Dr. Susana Guerra
Universidad Autónoma
Departament of Preventive Medicine and Public and Microbiology
Arzobispo Morcillo s/n
Madrid, Madrid 28049
Spain

Re: Spectrum03893-22R1 (ISG15 is a novel regulator of lipid metabolism during *Vaccinia virus* infection)

Dear Dr. Susana Guerra:

Your manuscript has been accepted, and I am forwarding it to the ASM Journals Department for publication. You will be notified when your proofs are ready to be viewed.

Sincerely,

Miguel Martinez
Editor, Microbiology Spectrum
